# A genetically encoded fluorescent biosensor for extracellular L-lactate

Yusuke Nasu[1], Ciaran Murphy-Royal [2,3], Yurong Wen[4,5], Jordan N. Haidey[2], Rosana S. Molina [6], Abhi Aggarwal[7], Shuce Zhang [8], Yuki Kamijo[1], Marie-Eve Paquet [9,10], Kaspar Podgorski [7], Mikhail Drobizhev [6], Jaideep S. Bains[2], M. Joanne Lemieux [4], Grant R. Gordon [2] & Robert E. Campbell [1,8 ✉]

L-Lactate, traditionally considered a metabolic waste product, is increasingly recognized as an important intercellular energy currency in mammals. To enable investigations of the emerging roles of intercellular shuttling of L-lactate, we now report an intensiometric green fluorescent genetically encoded biosensor for extracellular L-lactate. This biosensor, designated eLACCO1.1, enables cellular resolution imaging of extracellular L-lactate in cultured mammalian cells and brain tissue.

[1] Department of Chemistry, School of Science, The University of Tokyo, Bunkyo-ku, Tokyo 113-0033, Japan. [2] Hotchkiss Brain Institute, Cumming School of Medicine, Department of Physiology and Pharmacology, University of Calgary, Calgary, AB T2N 4N1, Canada. [3] Centre Hospitalier de l'Université de Montréal, Department of Neuroscience, Faculty of Medicine, University of Montreal, Montreal, QC H2X 0A9, Canada. [4] Department of Biochemistry, University of Alberta, Edmonton, AB T6G 2H7, Canada. [5] Department of Talent Highland, The First Affiliated Hospital, Xi'an Jiaotong University, Xi'an, Shaanxi 710061, China. [6] Department of Microbiology and Cell Biology, Montana State University, Bozeman, MT 59717, USA. [7] Janelia Research Campus, Howard Hughes Medical Institute, Ashburn, VA 20147, USA. [8] Department of Chemistry, University of Alberta, Edmonton, AB T6G 2G2, Canada. [9] CERVO Brain Research Center, Laval University, Québec, QC G1E 1T2, Canada. [10] Department of Biochemistry, Microbiology and Bioinformatics, Laval University, Québec, QC G1J 2G3, Canada. ✉email: campbell@chem.s.u-tokyo.ac.jp

Traditionally, the conjugate acid-base pair of L-lactic acid and L-lactate have been considered a "waste" by-product of glucose metabolism[1]. However, growing evidence suggests that L-lactate is better considered a crucial biological "fuel currency", that is shuttled from cell-to-cell[1]. For example, the astrocyte-to-neuron lactate shuttle (ANLS) hypothesis states that astrocytes metabolize glucose to produce L-lactate which is then released to the extracellular environment and taken up by neurons. In neurons, L-lactate is converted to pyruvate which enters the citric acid cycle to produce the energy necessary to sustain heightened neural activity[2]. The ANLS hypothesis remains controversial, with recent reports of evidence both for[2] and against[3] it.

Investigations of such cell-to-cell L-lactate shuttles would be facilitated by a genetically encoded fluorescent biosensor that would enable high resolution spatially and temporally resolved imaging of the extracellular L-lactate concentration. Genetically encoded fluorescent biosensors are powerful tools for cell-based and in vivo imaging of molecules, ions, and protein activities, and many examples have been reported to date[4,5]. However, despite remarkable progress in the development of such biosensors, including ones for intracellular L-lactate[6,7], no genetically encoded biosensors for extracellular L-lactate have yet been reported.

Here, we report the development of a genetically encoded fluorescent biosensor for extracellular L-lactate. This biosensor, designated eLACCO1.1, is the end-product of extensive directed evolution and structure-based mutagenesis followed by optimization of biosensor expression and localization on the cell surface. We confirm that eLACCO1.1 enables cellular resolution imaging of extracellular L-lactate in cultured mammalian cells and brain tissue.

## Results

### Development and characterization of a genetically encoded L-lactate biosensor, eLACCO1.

Periplasmic binding proteins (PBPs) derived from prokaryotic organisms have proven to be particularly effective sensing domain for extracellular single fluorescent protein-based biosensors[5], with key examples that include glutamate[8], GABA[9], acetylcholine[10], and serotonin[11]. To construct a prototype L-lactate biosensor, we inserted circularly permuted green fluorescent protein (cpGFP) into *Thermus thermophilis* TTHA0766 L-lactate binding periplasmic protein at 70 different positions (Supplementary Fig. 1). Positions of insertion sites in TTHA0766 were chosen by manual inspection of the protein crystal structure to identify loop regions that were solvent exposed and likely to undergo L-lactate-dependent conformational changes[5,12]. The variant (designated eLACCO0.1) with the largest change in fluorescence intensity ($\Delta F/F = (F_{max} - F_{min})/F_{min}$) upon L-lactate treatment had cpGFP inserted at position 191, and exhibited an inverse response (fluorescence decrease upon binding) with $\Delta F/F = 0.3$. Efforts to create prototype biosensors by inserting cpGFP into the analogous regions of TTHA0766 homologs produced no variants with larger L-lactate-dependent changes in fluorescence intensity (Supplementary Fig. 2). Accordingly, we focused our efforts on further development of eLACCO0.1.

To develop variants of eLACCO0.1 with larger $\Delta F/F$, we performed two rounds of linker optimization, followed by seven rounds of directed evolution by random mutagenesis of the whole gene, with screening for L-lactate-dependent change in fluorescence intensity (Fig. 1a). This effort ultimately produced the direct response variant eLACCO1 with $\Delta F/F$ of 6 (Fig. 1b–d and Supplementary Fig. 3) and a strict requirement for $Ca^{2+}$ at concentrations greater than 0.6 μM (Supplementary Fig. 4a, b). The apparent dissociation constant ($K_d$) of eLACCO1 is 4.1 μM

and 120 μM for L-lactate and D-lactate, respectively (Supplementary Fig. 4c). eLACCO1 showed a pH dependence similar to that of the GCaMP6f $Ca^{2+}$ biosensor, with $pK_a$ values of 6.0 and 8.7 in the presence and absence of L-lactate, respectively (Supplementary Fig. 4d)[13]. Relative to L-lactate, eLACCO1 was ~320, ~460, and ~1320 times less sensitive to the structurally similar molecules β-hydroxybutyrate, pyruvate and oxaloacetate, respectively (Supplementary Fig. 4c).

### The X-ray crystal structure of eLACCO1.

In an effort to obtain molecular insight into the structure and mechanism of eLACCO1, we determined the crystal structure of eLACCO1 in the L-lactate-bound state at a resolution of 2.25 Å (Fig. 1e and Supplementary Table 1). The overall structure reveals that the cpGFP-derived and TTHA0766-derived domains are closely associated via an extensive interaction surface that contains numerous molecular contacts. The TTHA0766-derived domain of eLACCO1 retains the same dimeric structure as TTHA0766 itself (Supplementary Fig. 5)[12]. The Hill coefficient of eLACCO1 is close to one, suggesting that the protomers in the dimer do not interact cooperatively (Supplementary Fig. 4c). Trp509 in the dimer interface forms a hydrophobic interaction with its symmetry-related self via π-π stacking. Mutagenesis of this residue abrogated L-lactate-dependent fluorescence presumably due to disruption of the dimeric structure, providing support for the conclusion that eLACCO1 functions as a dimer (Supplementary Fig. 5a).

### Development and characterization of affinity-tuned eLACCO1.1.

As the physiological concentration of extracellular L-lactate at rest is ~0.2 mM in the brain[14] and 1 mM in serum[15], we expected that the affinity of eLACCO1 for L-lactate (apparent $K_d$ ~4.1 μM) would be too high to respond to physiologically relevant concentration changes. To decrease the affinity for L-lactate, we introduced the structure-guided Tyr80Phe mutation to remove a hydrogen bond between L-lactate and eLACCO1. This produced the low-affinity eLACCO1.1 variant with apparent $K_d$ of 3.9 mM (Supplementary Fig. 6a, b). A non-responsive control biosensor, designated deLACCO, was engineered by incorporating the Asp444Asn mutation to abolish L-lactate binding (Supplementary Fig. 6c, d).

With the affinity-optimized eLACCO1.1 variant in hand, we undertook a detailed characterization of its spectral properties. eLACCO1.1 has two absorbance peaks at 397 and 496 nm, indicative of the neutral (protonated) and the anionic (deprotonated) chromophore, respectively (Fig. 2a). Consistent with the absorbance spectrum, eLACCO1.1 displays excitation peaks at 398 and 493 nm, and excitation at either peak produces an emission peak at 510 nm (Fig. 2b). To investigate the effect of $Ca^{2+}$ on the biosensor functionality, we determined the dependence of the fluorescence intensity of eLACCO1.1 on L-lactate and $Ca^{2+}$. These experiments revealed that the biosensor only functions as an L-lactate biosensor at concentrations of $Ca^{2+}$ greater than 9 μM (Fig. 2c). In vitro characterization of eLACCO1.1 revealed the brightness of the L-lactate-bound state to be ~80% and 42% of EGFP under one-photon and two-photon illumination, respectively (Table 1)[16,17]. eLACCO1.1 under one-photon excitation yields L-lactate-dependent decrease in excitation at the 398 nm peak and increase at the 493 nm peak (Fig. 2b). Under two-photon excitation (Fig. 2d), eLACCO1.1 also exhibited L-lactate-dependent ratiometric changes in excitation with an increase of the 924 nm peak and decrease of the 804 nm peak, corresponding to a shift from the neutral to the anionic state of the chromophore. The intensiometric L-lactate-induced two-photon-excited fluorescence change at 940–1000 nm ($\Delta F_2/F_2 = 8$–10) is about twice as large as the one-photon-excited

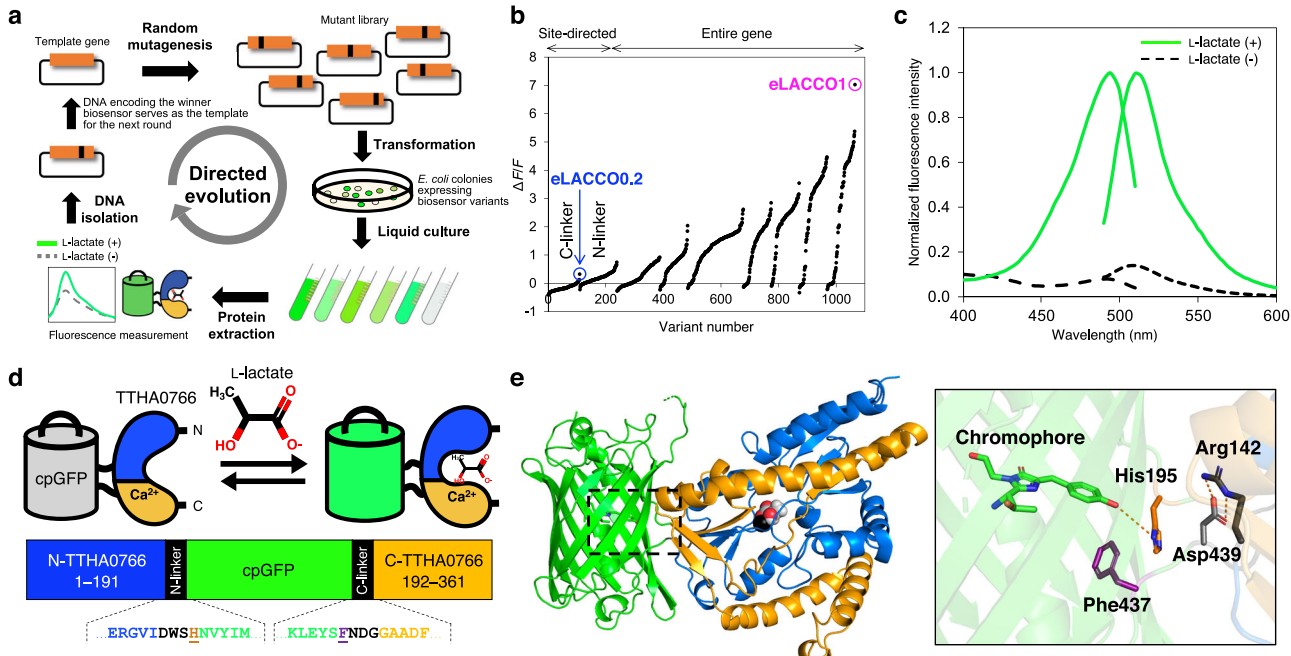

**Fig. 1 Development of a genetically encoded L-lactate biosensor, eLACCO1. a** Schematic of the directed evolution workflow. Specific sites (i.e., the linkers) or the entire gene of template L-lactate biosensor genes were randomly mutated and the resulting mutant library was used to transform *E. coli*. Bright colonies were picked and cultured, and then proteins were extracted and $\Delta F/F$ upon addition of 10 mM L-lactate was determined. The variant with the highest $\Delta F/F$ was used as the template for the next round. **b** $\Delta F/F$ rank plot representing all proteins tested during the directed evolution. For each round, tested variants are ranked from lowest (negative responses indicate inverse response) to highest $\Delta F/F$ value from left to right. The first round of evolution used a library in which residues of the C-terminal linker were randomized. Screening of this library led to the discovery of eLACCO0.2 with a direct response (fluorescence increase upon binding) to L-lactate ($\Delta F/F = 0.3$) while the template eLACCO0.1 decreased the fluorescence intensity (inverse response) in response to L-lactate. Nine rounds of the evolution led to eLACCO1 indicated with a magenta circle. **c** Excitation and emission spectra of eLACCO1 in the presence and absence of 10 mM L-lactate. Excitation and emission peak is at 494 and 512 nm, respectively. **d** Schematic representation of eLACCO1 and its mechanism of response to L-lactate. Linker regions are shown in black and the two "gate post" residues[5] in cpGFP are highlighted in dark orange (His195) and purple (Phe437). **e** Crystal structure of eLACCO1. Right panel represents a zoom-in view around the chromophore. Source data of **b**, **c** are provided as a Source Data file.

fluorescence change ($\Delta F/F = 4$, at 480–510 nm) (Fig. 2b, d). A similar trend has been observed for some red fluorescent genetically encoded $Ca^{2+}$ biosensors[18]. The fluorescence of eLACCO1.1 is pH dependent, exhibiting $pK_a$ values of 7.4 and 9.4 in the presence and absence of L-lactate, respectively (Fig. 2e). The control biosensor deLACCO showed no response to L-lactate, and pH dependence that was similar to the lactate-free state of eLACCO1.1 (Supplementary Fig. 7). Investigation of the molecular specificity of eLACCO1.1 revealed that the decreased binding affinity extended to structurally similar molecules, with pyruvate and oxaloacetate causing negligible fluorescent response even at 100 mM (Fig. 2f). eLACCO1.1 responds to D-lactate with an apparent $K_d$ of 21 mM, a concentration that is far greater than the ~11–70 nM concentration in serum[19]. Overall, these results indicate that eLACCO1.1 is likely to respond only to changes in L-lactate concentration and pH under physiological conditions.

**Targeting of eLACCO1.1 to the extracellular environment.** To test eLACCO1.1 as a genetically encoded fluorescent biosensor for extracellular L-lactate in a cellular milieu, we targeted it to the surface of mammalian cells by fusing it to various N-terminal leader sequences and C-terminal anchor domains (Fig. 3a). The widely used combination of the immunoglobulin κ-chain (Igκ) leader sequence and the platelet-derived growth factor receptor (PDGFR) transmembrane domain resulted in only intracellular expression with localization reminiscent of the nuclear membrane (Fig. 3b). We screened a range of leader sequences in combination with the PDGFR anchor, but did not discover any that led to robust

membrane localization of PDGFR-anchored eLACCO1.1 (Supplementary Fig. 8). Hence, we turned to the combination of an N-terminal leader sequence and a glycosylphosphatidylinositol (GPI) anchor, both of which are derived from CD59. Ultimately, we found that the combination of a CD59-derived N-terminal leader sequence, and a CD59-derived GPI anchor, provided the desired targeting of eLACCO1.1 to the cell surface (Fig. 3b). To further optimize the biosensor functionality, we screened a number of linkers between eLACCO1.1 and GPI anchor. This experiment revealed that the 18 amino acid linker (GSTSGSGKPGSGEGSTKG) provides the best $\Delta F/F$ upon treatment with L-lactate (Fig. 3c, d). Bath application of 10 mM L-lactate resulted in no significant difference in $\Delta F/F$ in the presence versus absence of monocarboxylic transporter inhibitor AR-C155858, though the variability in $\Delta F/F$ was greatly decreased by this treatment (Fig. 3e). This result suggests that only the fraction of eLACCO1.1 on the cell surface contributes to the fluorescence response associated with changes in the extracellular L-lactate concentration. The decreased variability associated with AR-C155858 treatment is attributed greater consistency in extracellular lactate concentration due to blockage of lactate uptake into cells.

**Characterization of eLACCO1.1 in live mammalian cells.** We characterized cell-surface-targeted eLACCO1.1 in terms of several important parameters. eLACCO1.1, with an optimized linker between eLACCO1.1 and GPI anchor, robustly increased fluorescence intensity ($\Delta F/F$ of $3.3 \pm 0.3$, mean ± s.e.m., $n = 26$ cells) upon treatment with L-lactate (Fig. 4a, b). The control biosensor

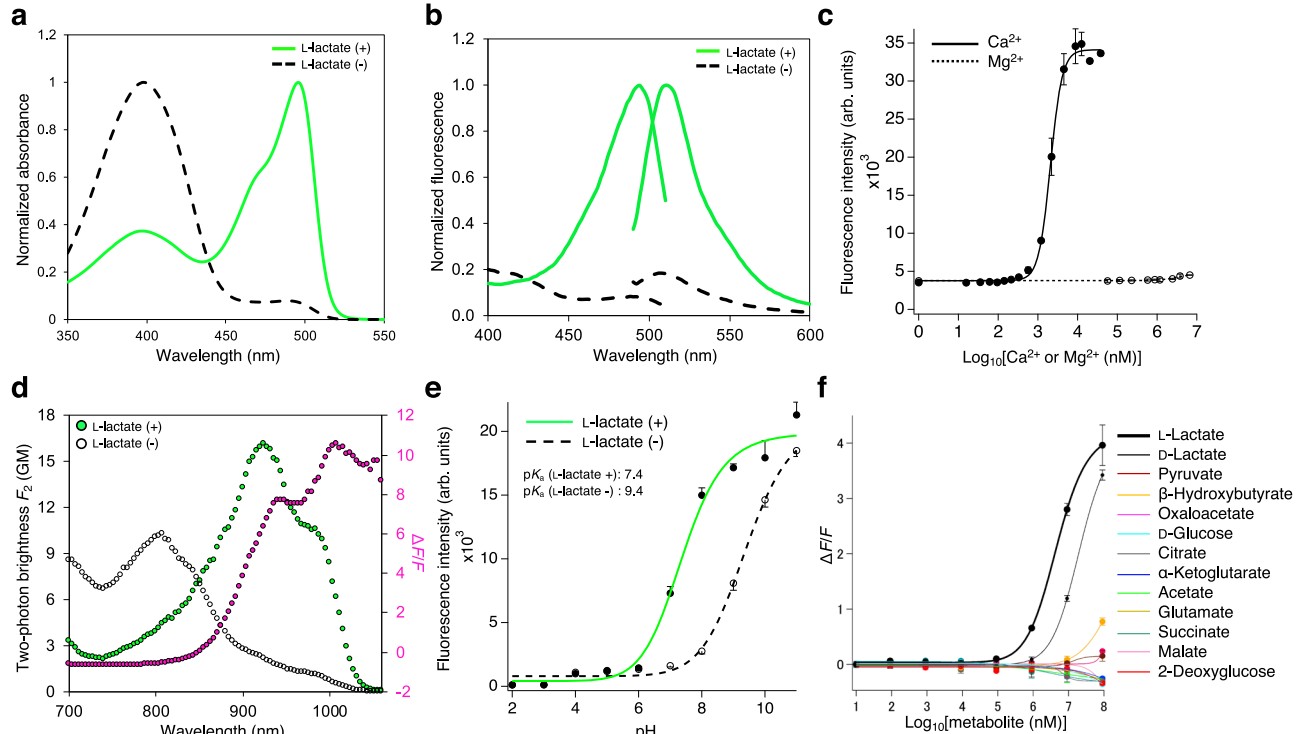

**Fig. 2 In vitro characterization of affinity-tuned eLACCO1.1. a** Absorbance spectra of eLACCO1.1 in the presence (100 mM) and absence of L-lactate. **b** Excitation and emission spectra of eLACCO1.1 in the presence (100 mM) and absence of L-lactate. **c** Fluorescence of eLACCO1.1 in the presence of 100 mM L-lactate as a function of $Ca^{2+}$ and $Mg^{2+}$. $N = 3$ independent experiments (mean ± s.d.). **d** Two-photon excitation spectra of eLACCO1.1 in the presence (100 mM) and absence of L-lactate. $\Delta F/F$ is represented in magenta. GM Goeppert-Mayer units. **e** pH titration curves of eLACCO1.1 in the presence and absence of 100 mM L-lactate. $n = 3$ independent experiments (mean ± s.d.). **f** Dose-response curves of eLACCO1.1 for L-lactate and a variety of metabolites. $n = 3$ independent experiments (mean ± s.d.). Source data are provided as a Source Data file.

**Table 1 One- and two-photon photophysical parameters of eLACCO1.1.**

| | eLACCO1.1 | | EGFP |
|---|---|---|---|
| | L-lactate free | L-lactate bound | |
| Relative fraction of neutral chromophore ($\rho_N$) | 0.97 ± 0.07 | 0.57 ± 0.04 | 0.07[a] |
| Relative fraction of anionic chromophore ($\rho_A$) | N/A | 0.43 ± 0.03 | 0.93[a] |
| Neutral extinction coefficient (mM$^{-1}$ cm$^{-1}$, $\varepsilon_N$) | 34 | 38 | N/A |
| Anionic extinction coefficient (mM$^{-1}$ cm$^{-1}$, $\varepsilon_A$) | 78 | 89 | 56.0[b]; 58.3[c] |
| Neutral quantum yield ($\varphi_N$) | 0.20 ± 0.01 | 0.23 ± 0.01 | N/A |
| Anionic quantum yield ($\varphi_A$) | 0.60 ± 0.03 | 0.78 ± 0.04 | 0.67[b]; 0.76[c] |
| Molecular brightness ($\rho_A \times \varepsilon_A \times \varphi_A$) | N/A | 29.9 | 34.9[b]; 41.2[c] |
| Neutral 1PA peak (nm) | 398 | 397 | N/A |
| Anionic 1PA peak (nm) | 493 | 496 | 488[b] |
| Two-photon brightness $F_2$ (GM)[d] | 2.2 ± 0.3 (924 nm) | 16 ± 2 (924 nm) | 38 (911 nm)[c] |

Mean ± s.d. N/A, not applicable. Subscript A and N correspond to the anionic and neutral form of chromophore, respectively.
[a]Calculated using the equation, $\log_{10}([anionic]/[neutral]) = pH - pK_a$, where pH is 7.2 and $pK_a$ is 6.1 (ref. [16]).
[b]Data from ref. [16].
[c]Data from ref. [17].
[d]Two-photon brightness is calculated as $F_2 = \rho_A \times \sigma_{2,A} \times \varphi_A$ at the peak wavelength (in parentheses), where $\sigma_{2,A}$ is the peak two-photon absorption cross section.

deLACCO had similar membrane localization and, as expected, did not respond to L-lactate (Fig. 4a, b). eLACCO1.1 also exhibited an L-lactate-dependent change in the ratio of excitation at 365 and 470 nm, suggesting eLACCO1.1 could be applicable as both an intensiometric and a ratiometric biosensor (Supplementary Fig. 9). To test photostability, we continuously illuminated eLACCO1.1-expressing cells using one-photon wide-field microscopy (Fig. 4c). eLACCO1.1 showed photostability that is comparable to EGFP and cpGFP. To examine the on-rate kinetics of eLACCO1.1, we bathed eLACCO1.1-expressing HeLa cells in a solution containing 10 mM L-lactate. L-Lactate application induced the robust increase in the fluorescence with the on rate ($\tau_{on}$) of 1.2 min (Fig. 4d). Cell-surface-targeted eLACCO1.1 has an in situ apparent $K_d$ of 1.6 mM (Fig. 4e) and displays $Ca^{2+}$ and pH dependent fluorescence as shown in Supplementary Figs. 10 and 11. Stopped-flow analysis of purified eLACCO1.1 protein revealed that the off-rate kinetics are faster than the 1.1 ms dead time of the instrument used (Supplementary Fig. 12). To characterize the performance of eLACCO1.1 in neurons, we expressed eLACCO1.1 in rat primary cortical neurons. We observed that neurons expressing eLACCO1.1 exhibited bright membrane-localized fluorescence with some puncta apparent (Fig. 4f). On

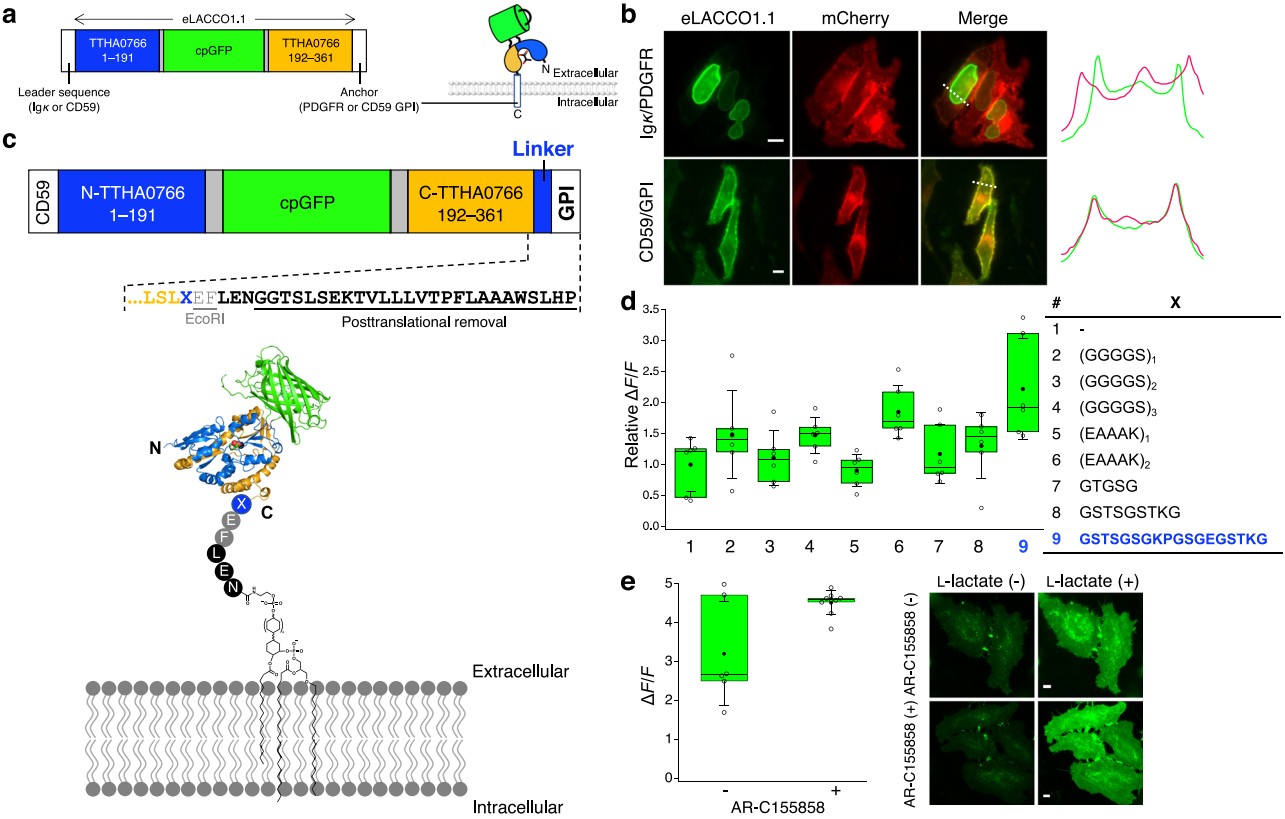

**Fig. 3 Targeting of eLACCO1.1 to the extracellular environment. a** Schematic representation of the domain structure of eLACCO1.1 with N-terminal leader sequence and C-terminal anchor domain. **b** Localization of eLACCO1.1 with different leader sequences and anchors in HeLa cells. mCherry red fluorescent protein (RFP) with Igκ leader sequence and PDGFR transmembrane domain was used as cell surface marker. Line-scans (right) correspond to dashed white lines on the merged images. eLACCO1.1 with CD59-derived leader and anchor domain is expressed on cell surface as indicated by colocalization with cell-surface-targeted mCherry. Similar results were observed in more than 10 cells. Scale bars, 10 μm. **c** Schematic representations of the eLACCO1.1 on cell surface with CD59 leader sequence and CD59 GPI anchor. Underlined sequence is removed after the translation, followed by GPI labeling to the C-terminal Asn (N) residue. **d** Relative $\Delta F/F$ of a range of eLACCO1.1 linker variants. The linker GSTSGSGKPGSGEGSTKG has been demonstrated to reduce aggregation and be resistant to proteolysis[47]. $n = 6$ cells for each linker variant. **e** $\Delta F/F$ of the linker-optimized eLACCO1.1 in the presence and absence of 1 μM AR-C155858. $n = 6$ and 9 cells for AR-C155858($-$) and AR-C155858($+$), respectively. Two-tailed student's *t*-test with Welch's correction, $P > 0.05$. Scale bars, 10 μm. In the box plots in **d**, **e**, the horizontal line is the median; the top and bottom horizontal lines are the 25th and 75th percentiles for the data; and the whiskers extend one standard deviation range from the mean represented as black filled circle. Source data of **d**, **e** are provided as a Source Data file.

the surface of rat primary cortical neurons, eLACCO1.1 displayed a $\Delta F/F$ of $2.0 \pm 0.3$ upon bath application of 10 mM L-lactate (mean ± s.e.m., $n > 10$ neurons from 3 cultures, Fig. 4f). Attempts to use the previously reported Förster resonance energy transfer (FRET)-based biosensor Laconic[6] for imaging of extracellular L-lactate were unsuccessful (Fig. 4b and Supplementary Fig. 13). Taken together, these results indicated that eLACCO1.1 could uniquely be useful for imaging of extracellular L-lactate concentration dynamics.

**Two-photon imaging of L-lactate on astrocytes in acute brain slices**. The ANLS hypothesis states that glial cells such as astrocytes can release L-lactate into the extracellular space and that this L-lactate is taken up by neurons to serve as an energy source[2]. To determine whether eLACCO1.1 remains functional on the surface of astrocytes of mammalian brain tissue, we used two-photon microscopy to examine cortical acute brain slices prepared from mice injected with an adeno-associated virus (AAV) coding eLACCO1.1 under the control of an astrocyte-specific promoter GFAP (Fig. 5a). Bath application of L-lactate elicited a variable and significant increase in eLACCO1.1 fluorescence at all doses tested: 1 mM ($\Delta F/F = 0.19 \pm 0.04$), 2.5 mM ($\Delta F/F = 0.48 \pm 0.15$), and 10 mM ($\Delta F/F = 0.65 \pm 0.17$) (Fig. 5b–d). Collectively, these ex vivo data indicate that eLACCO1.1 enables detection of extracellular

L-lactate in acute brain slice and could potentially be applicable to imaging the release of L-lactate from astrocytes in an ex vivo brain preparation.

**Imaging of endogenous L-lactate release from cultured glioblastoma cells**. To determine if eLACCO1.1 can enable imaging of the release of endogenous L-lactate from cells, we targeted eLACCO1.1 to the surface of T98G glioma cells. Upon treatment with a high glucose concentration (25 mM), which is expected to stimulate the production of endogenous L-lactate, eLACCO1.1 on the surface of T98G cells underwent an increase in fluorescence consistent with increased secretion of L-lactate (Fig. 6a–c). In the presence of phloretin or AR-C155858, two inhibitors of the monocarboxylate transporter, the glucose-dependent fluorescence increase was diminished. In both the presence and absence of phloretin or AR-C155858, the control biosensor deLACCO showed no substantial change in fluorescence intensity, indicating that the observed fluorescence changes were due to the L-lactate-dependent response of eLACCO1.1. To image the production and export of endogenous L-lactate under a more physiologically relevant condition, we observed eLACCO1.1-expressing T98G cells treated with a physiological plasma concentration of glucose (5.6 mM, Fig. 6d)[20]. We first attempted to inhibit production of endogenous L-lactate by treating the cells with 100 μM

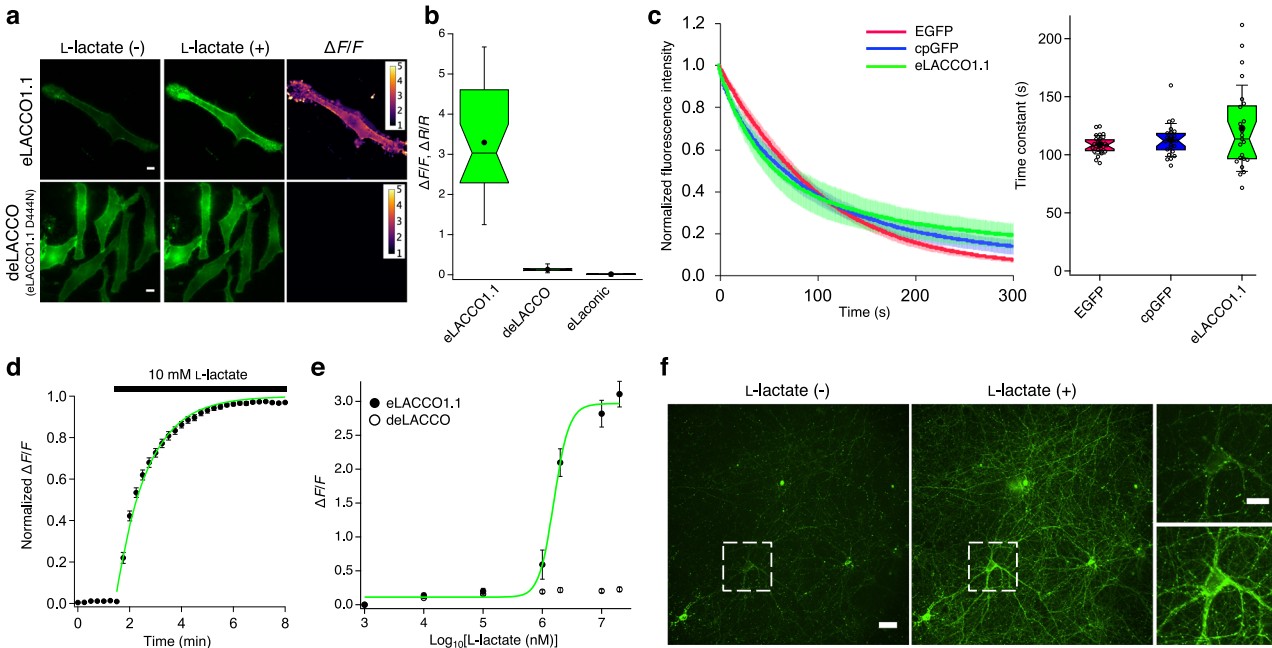

**Fig. 4 Characterization of eLACCO1.1 in live mammalian cells. a** Representative images of eLACCO1.1 and control biosensor deLACCO expressed on cell surface of HeLa cells before and after 10 mM L-lactate stimulation. Similar results were observed in more than 10 cells. Scale bars, 10 μm. **b** Box plots of L-lactate-dependent $\Delta F/F$ for eLACCO1.1 and deLACCO, and $\Delta R/R$ for eLaconic. $n = 26$, 28 and 13 cells for eLACCO1.1, deLACCO and eLaconic, respectively. **c** Photobleaching curves (left) and time constants (right) for EGFP, cpGFP and eLACCO1.1 expressed on the cell surface. To test photostability, we continuously illuminated eLACCO1.1-expressing cells using one-photon wide-field microscopy. $n = 25$, 24 and 23 cells for EGFP, cpGFP and eLACCO1.1, respectively. In the photobleaching curves, solid lines represent mean value and shaded area represent s.d. One-way ANOVA, $P > 0.05$. **d** Time-course of the fluorescence response of eLACCO1.1 on cell surface under L-lactate stimulation. To examine the on-rate kinetics of eLACCO1.1, we bathed eLACCO1.1-expressing HeLa cells in a solution containing 10 mM L-lactate. The plot was fitted with a single exponential function. $n = 13$ cells (mean ± s.e.m.). **e** In situ titration of L-lactate on HeLa cells. Data were fitted with Hill equation. $n = 11$ cells (mean ± s.e.m.). **f** Representative images of eLACCO1.1 expressed in primary cortical neurons before and after 10 mM L-lactate stimulation. eLACCO1.1 was expressed at relatively high levels, as indicated by bright membrane-localized fluorescence with some puncta apparent. Similar results were observed in more than 10 neurons from three independent experiments. Scale bars, 10 μm. In the box plots in **b**, **c**, the narrow part of notch with the horizontal line is the median; the top and bottom of the notch denote the 95% confidence interval of the median; the top and bottom horizontal lines are the 25th and 75th percentiles for the data; and the whiskers extend one standard deviation range from the mean represented as black filled circle. Source data of **b**–**e** are provided as a Source Data file.

iodoacetate, an inhibitor of glyceraldehyde 3-phosphate dehydrogenase. Unexpectedly, this stimulation resulted in apparent aggregation of eLACCO1.1 (Supplementary Fig. 14). To avoid this iodoacetate-induced aggregation artefact, we turned to using NCI-737, an inhibitor of lactate dehydrogenase (LDH) (Fig. 6e, f). Treatment with NCI-737 caused a slight increase in the fluorescence response of deLACCO. In contrast, eLACCO1.1 showed a decrease in the fluorescence response upon NCI-737 treatment. The opposite effects of NCI-737 on the responses of eLACCO1.1 and deLACCO are consistent with the observed fluorescence changes being due to changes in the extracellular L-lactate concentration. Overall, these results demonstrate that eLACCO1.1 enables imaging of extracellular L-lactate release from glial cells with cellular resolution.

## Discussion

This study describes the development of a genetically encoded extracellular L-lactate biosensor, designated eLACCO1.1. Screening of a library of variants with cpGFP inserted into different positions of *Thermus thermophilis* TTHA0766 L-lactate binding protein led to the identification of a biosensor prototype in which the L-lactate-dependent conformational change of TTHA0766 allosterically modulates the fluorescence intensity of cpGFP. Extensive directed evolution of the prototype led to the high-performance L-lactate biosensor, eLACCO1. Rational mutagenesis

based on the crystal structure of eLACCO1 tuned its L-lactate affinity to be optimal for the physiological concentration range of extracellular L-lactate. An intensive effort to target the affinity-tuned variant to the extracellular environment eventually produced eLACCO1.1 that enables cellular resolution imaging of extracellular L-lactate in cultured mammalian cells and brain tissue.

The choice of sensing domain is critically important to the development of any genetically encoded biosensor[5]. Biosensors for extracellular applications (e.g., those with specificity for neurotransmitters like glutamate[8], GABA[9], acetylcholine[10,21], serotonin[11,22,23], dopamine[24,25], and norepinephrine[26]) have generally used microbial PBPs or G-protein-coupled receptors (GPCRs) as the sensing domain. In this work, we chose to use *Thermus thermophilis* TTHA0766 L-lactate-binding PBP. Relative to other possible sensing domains that are normally found in the reducing environment of the cytoplasm, an advantage of microbial PBPs is that they naturally function in the oxidative environment of the periplasm. This property is likely beneficial to PBP-based biosensors for extracellular targets, since the protein must be trafficked through oxidative organelles such as the endoplasmic reticulum and Golgi apparatus and ultimately exposed to the oxidative extracellular environment. A gluconate operon repressor family transcription factor, LldR, has previously been used as the sensing domain for intracellular L-lactate biosensors[6,7]. Attempts to use the Laconic LldR-based biosensor

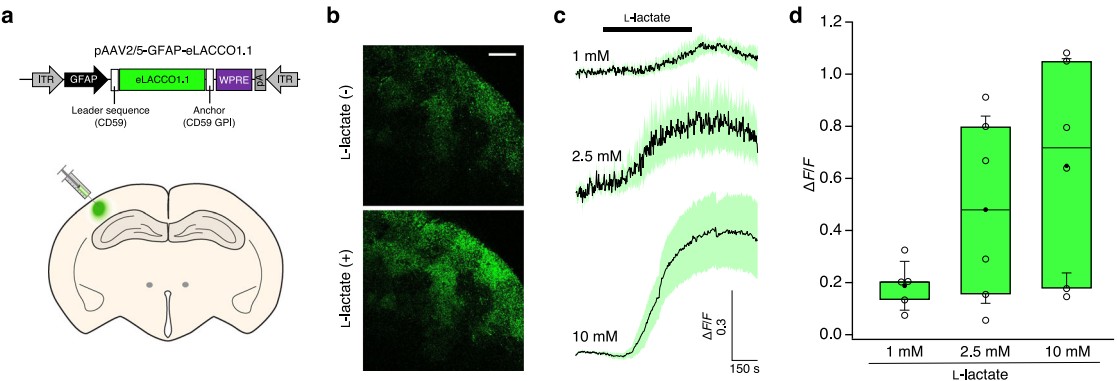

**Fig. 5 Two-photon imaging of ʟ-lactate on astrocytes in acute brain slices. a** Schematic illustration of AAV injection into the somatosensory cortex for the brain slice experiments. ITR inverted terminal repeat, GFAP human glial fibrillary acidic protein promoter, WPRE woodchuck hepatitis virus posttranslational regulatory element, pA human growth hormone polyA signal. **b** Representative two-photon images of eLACCO1.1 expressed on astrocytes in brain slice before and after 10 mM ʟ-lactate stimulation. Parallel experiments with deLACCO produced baseline fluorescence that was not significantly higher than background and did not produce a significant change in fluorescence relative to baseline ($\Delta F/F = 0.85 \pm 1.6$, $P = 0.25$, two-tailed paired $t$-test). Similar results were observed from more than 3 slices. Scale bar represents 50 μm. **c** Fluorescence traces of eLACCO1.1-expressing astrocytes in response to bath application of ʟ-lactate (mean ± s.e.m.). 1 mM ʟ-lactate ($n = 5$ slices from 4 mice), 2.5 mM ʟ-lactate ($n = 5$ slices from 4 mice), or 10 mM ʟ-lactate ($n = 6$ slices from 4 mice). **d** $\Delta F/F$ plots for eLACCO1.1 at each dose of ʟ-lactate. $\Delta F/F$ was calculated by: $\Delta F/F = ((F_x - F_b)/F_b)$, where $F_x$ is the peak fluorescence intensity and $F_b$ is the baseline fluorescence intensity. 1 mM ʟ-lactate ($n = 5$ slices from 4 mice, $P = 0.01$, two-tailed paired $t$-test), 2.5 mM ʟ-lactate ($n = 6$ slices from 4 mice, $P = 0.02$, two-tailed paired $t$-test), or 10 mM ʟ-lactate ($n = 6$ slices from 4 mice, $P = 0.01$, two-tailed paired $t$-test). The horizontal line is the median; the top and bottom horizontal lines are the 25th and 75th percentiles for the data; and the whiskers extend one standard deviation range from the mean represented as black filled circle. One slice data in the 2.5 mM group was omitted from the trace in **c** because it was not a complete time course, but $\Delta F/F$ of the slice could be measured and was included in **d**. Source data of **c**, **d** are provided as a Source Data file.

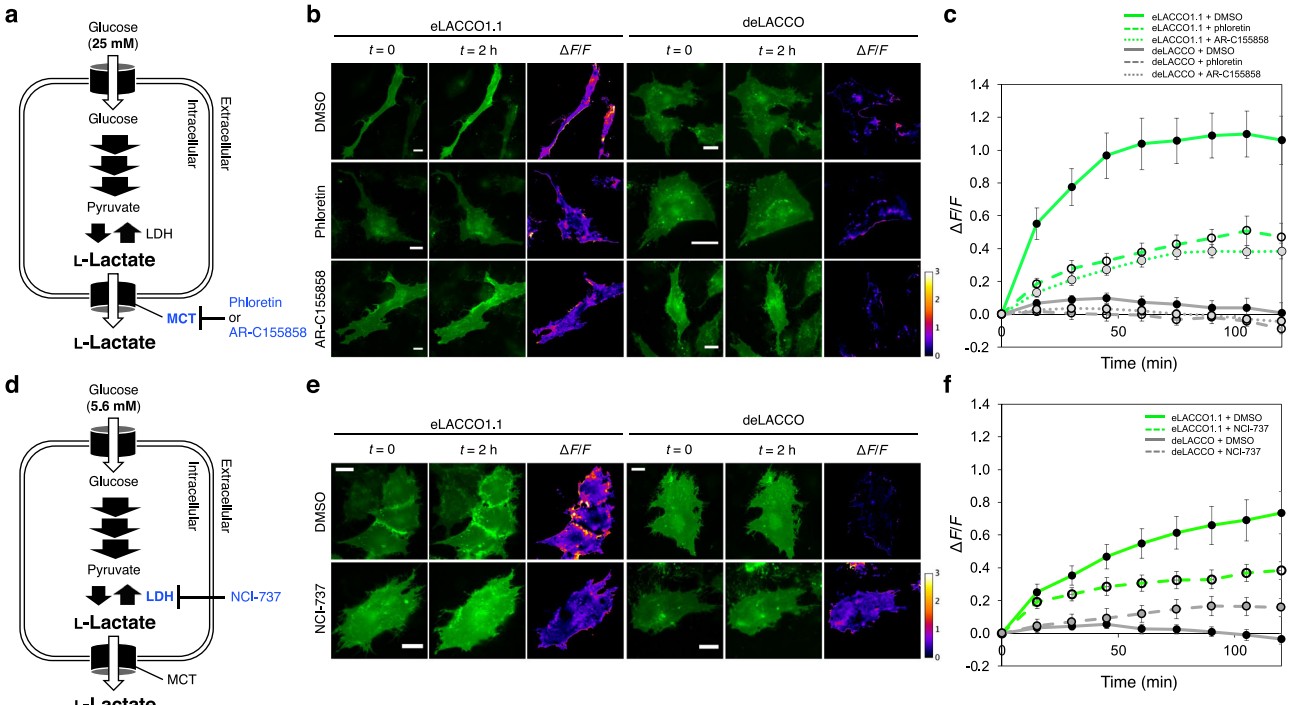

**Fig. 6 Imaging of endogenous ʟ-lactate release from glioblastoma cells. a** Schematic representation of ʟ-lactate metabolism in the presence of 25 mM of extracellular glucose. MCT monocarboxylate transporter, LDH lactate dehydrogenase. **b** Representative fluorescence images of T98G cells expressing eLACCO1.1 or deLACCO treated with 25 mM glucose. Scale bars, 20 μm. **c** $\Delta F/F$ versus time for T98G cells expressing eLACCO1.1 or deLACCO. $n = 7$ and 8 cells for eLACCO1.1 and deLACCO, respectively (mean ± s.e.m.). **d** Schematic representation of ʟ-lactate metabolism in the presence of 5.6 mM of extracellular glucose. **e** Representative fluorescence images of T98G cells expressing eLACCO1.1 or deLACCO treated with 5.6 mM glucose. Scale bars, 20 μm. **f** $\Delta F/F$ versus time for T98G cells expressing eLACCO1.1 or deLACCO. $n = 8$ and 8 cells for eLACCO1.1 and deLACCO, respectively (mean ± s.e.m.). Source data of **c**, **f** are provided as a Source Data file.

for imaging of extracellular L-lactate were unsuccessful (Fig. 4b and Supplementary Fig. 13), consistent with the suggested importance of using sensing domains that are compatible with the extracellular environment.

Relative to GPCR-based biosensors, there are several technical advantages associated with development of PBP-based single FP-based biosensors. Of particular relevance to the methods used in this work, directed evolution with bacterial expression is applicable to PBP-based biosensors which can be expressed in soluble form, but is not applicable to integral membrane proteins such as GPCRs. Furthermore, these soluble biosensors can often be crystallized in order to resolve their structure at atomic resolution, providing insights that guide and accelerate engineering efforts. These insights ultimately help to produce biosensors with fluorescent responses that are substantially larger than those yet demonstrated for GPCR-based biosensors.

The mechanism of eLACCO1.1 must involve changes in the chromophore environment that are induced by conformational changes that accompany binding of L-lactate to the TTHA0766-derived domain[5]. Based on the pH dependence, these changes in the chromophore environment are stabilizing the brightly fluorescent deprotonated form of the chromophore. The structure of eLACCO1, which we assume to be representative of eLACCO1.1, reveals that the imidazole side chain of His195 is likely a key moiety for mediating changes in the chromophore protonation state (Fig. 1e). Further insight into the mechanism comes from examining the position of beneficial mutations discovered during biosensor optimization. In the first round of the directed evolution for optimization of the C-terminal linker, the introduction of the Asn439Asp mutation converted the biosensor from having an inverse response to having a direct response (Fig. 1b). Accordingly, we propose that the mechanism of eLACCO1.1 involves a conformational switch between two states: the L-lactate-free state where the protonated (dark) chromophore is stabilized by interaction with carboxylate side chain of Asp439 (and His195 is further away); and the L-lactate-bound state where the deprotonated chromophore (bright) is stabilized by interaction with the imidazole side chain of His195 (and Asp439 is further away). Consistent with this proposed mechanism, in the final round of the evolution, we discovered the Lys142Arg mutation that substantially improved $\Delta F/F$ (Fig. 1b). The eLACCO1 crystal structure reveals that Arg142 forms a salt bridge with Asp439 in the L-lactate-bound state. This salt-bridge may further limit residual interaction of Asp439 with the chromophore (possibly due to increased distance or more effective charge neutralization) and thereby contribute to increased brightness of the L-lactate-bound state and a higher $\Delta F/F$.

In vitro characterization revealed that lactate-free eLACCO1.1 harbors 97% of the neutral (dark) chromophore and 3% of the anionic (bright) chromophore, and that this changes to 57% neutral and 43% anionic for lactate-bound eLACCO1.1 (Table 1). This chromophore equilibrium indicates that with respect to further improving the $\Delta F/F$, there is relatively little room to engineer decreased brightness of eLACCO1.1 in the lactate-free state. In contrast, the brightness of the lactate-bound state could be further engineered and improved, in principle, to 230% of its current value (i.e., 0% neutral and 100% anionic). Notably, the $\Delta F/F$ rank plot of directed evolution did not plateau (Fig. 1b), suggesting that additional rounds of directed evolution could yet improve $\Delta F/F$ by increasing the brightness of lactate-bound state. Considering that only 43% of the biosensor is present in the anionic form, the fluorescence brightness per one anionic chromophore is ~70% larger than that of EGFP[16]. This enhancement can be explained by the fact that the anionic form of eLACCO1.1 in the L-lactate-bound state has the extinction coefficient ~60% higher than EGFP. Excitation of the neutral form at 398 nm

results in fluorescence from the anionic excited state (at 510 nm), presumably due to excited-state proton transfer[27]. The intrinsic two-photon brightness ($F_2 = \sigma_2 \times \varphi \times \rho$, where $\sigma_2$ is the two-photon absorption cross section) of L-lactate-bound eLACCO1.1 at 924 nm ($F_2 = 16$ GM) is 42% that of EGFP ($F_2 = 38$ GM)[17]. This is because only 43% of eLACCO1.1 is present in the anionic form when bound to L-lactate. Though moderately dimmer than GFP in terms of two-photon brightness, it is notable that eLACCO1.1 in the L-lactate-bound state is 2.4-fold brighter than Citrine ($F_2 = 6.7$ GM)[28].

As with many genetically encoded biosensors[8,11], the maximal fluorescence response of eLACCO1.1 in acute brain slice ($\Delta F/F$ ~0.7) is smaller than that in HeLa cells ($\Delta F/F$ ~3) and cultured neurons ($\Delta F/F$ ~2). A previous study with the extracellular neurotransmitter serotonin biosensor iSeroSnFR similarly reported that its fluorescence response in acute brain slice ($\Delta F/F$ ~0.8) is smaller than that in HEK293T cells ($\Delta F/F$ ~8) and cultured neurons ($\Delta F/F$ ~6) upon treatment with 1 mM serotonin[11]. In addition to the smaller fluorescence response, eLACCO1.1 shows a relatively large variation of fluorescence response in acute brain slice (Fig. 5d). To develop a next generation eLACCO variant with improved performance in brain tissues, it might be necessary to combine bacteria-based high-throughput directed evolution with secondary neuron- or slice-based assessments to identify those variants that best retain their performance in brain tissues.

To date, the concentration and dynamics of extracellular L-lactate in brain tissue has typically been assessed using inserted enzyme-mediated electrodes[14,29]. Relative to electrodes, the inherent advantages of eLACCO1.1 are that it can be non-invasively introduced in the form of its corresponding gene, and it enables spatially-resolved imaging of L-lactate concentration dynamics. In addition, the targeted expression of eLACCO1.1 in specific cell types (e.g., astrocytes or neurons), will enable researchers to accurately determine which cell types are importing, and which are exporting, L-lactate. Accordingly, we anticipate that eLACCO1.1, and further improved variants, will play a central role in investigations of L-lactate shuttles, including the controversial ANLS hypothesis.

## Methods

**General methods and materials.** Synthetic DNA encoding the lactate binding bacterial periplasmic protein TTHA0766 was purchased from Integrated DNA Technologies. Phusion high-fidelity DNA polymerase (Thermo Fisher Scientific) was used for routine polymerase chain reaction (PCR) amplifications, and Taq DNA polymerase (New England Biolabs) was used for error-prone PCR. The QuikChange mutagenesis kit (Agilent Technologies) was used for site-directed mutagenesis. Restriction endonucleases, rapid DNA ligation kits and GeneJET miniprep kits were purchased from Thermo Fisher Scientific. PCR products and products of restriction digests were purified using agarose gel electrophoresis and the GeneJET gel extraction kit (Thermo Fisher Scientific). DNA sequences were analyzed by DNA sequence service of the University of Alberta Molecular Biology Service Unit and Fasmac Co., Ltd. Fluorescence excitation and emission spectra were recorded on Safire2 and Spark plate readers (Tecan).

**Engineering of eLACCO1.1.** The gene encoding cpGFP with N- and C- terminal linkers (LV and NP, respectively) was amplified using iGluSnFR gene as template, followed by insertion into each site of TTHA0766 lactate binding protein in a pBAD vector (Life Technologies) by Gibson assembly (New England Biolabs). Variants were expressed in *E. coli* strain DH10B (Thermo Fisher Scientific) in LB media supplemented with 100 μg mL$^{-1}$ ampicillin and 0.02% L-arabinose. Proteins were extracted using B-PER bacterial protein extraction reagent (Thermo Fisher Scientific) and tested for fluorescence brightness and lactate-dependent response. The most promising variant, designated as eLACCO0.1, was subjected to an iterative process of library generation and screening in *E. coli*. Libraries were generated by site-directed mutagenesis using QuikChange (Agilent Technologies) or error-prone PCR of the whole gene. For each round, ~100–200 fluorescent colonies were picked, cultured and tested on 384-well plates under a plate reader. There were 9 rounds of screening before eLACCO1 was identified. Finally, Tyr80Phe mutation was added to eLACCO1 to tune the lactate affinity using Q5 high-fidelity DNA polymerase (New England Biolabs). The resulting mutant was designated as eLACCO1.1.

**Protein purification and in vitro characterization**. The gene encoding eLACCO1.1, with a poly-histidine tag on the N-terminus, was expressed from the pBAD vector. Bacteria were lysed with a cell disruptor (Branson) and then centrifuged at 15,000 g for 30 min, and proteins were purified by Ni-NTA affinity chromatography (Agarose Bead Technologies). Absorption spectra of the samples were collected with a Lambda950 Spectrophotometer (PerkinElmer). To perform pH titrations, protein solutions were diluted into buffers (pH from 2 to 11) containing 30 mM trisodium citrate, 30 mM sodium borate, 30 mM MOPS, 100 mM KCl, 10 mM CaEGTA, and either no L-lactate or 10 mM L-lactate. Fluorescence intensities as a function of pH were then fitted by a sigmoidal binding function to determine the $pK_a$. For lactate titration, buffers were prepared by mixing an L-lactate ($-$) buffer (30 mM MOPS, 100 mM KCl, 1 mM $CaCl_2$, pH 7.2) and an L-lactate ($+$) buffer (30 mM MOPS, 100 mM KCl, 1 mM $CaCl_2$, 100 mM L-lactate, pH 7.2) to provide L-lactate concentrations ranging from 0 to 100 mM at 25 °C. Fluorescence intensities were plotted against L-lactate concentrations and fitted by a sigmoidal binding function to determine the Hill coefficient and apparent $K_d$. For $Ca^{2+}$ titration, buffers were prepared by mixing a $Ca^{2+}$ ($-$) buffer (30 mM MOPS, 100 mM KCl, 10 mM EGTA, 100 mM L-lactate, pH 7.2) and a $Ca^{2+}$ ($+$) buffer (30 mM MOPS, 100 mM KCl, 10 mM CaEGTA, 100 mM L-lactate, pH 7.2) to provide $Ca^{2+}$ concentrations ranging from 0 to 39 μM at 25 °C.

Rapid kinetic measurements for the interaction of eLACCO1 or eLACCO1.1 with L-lactate were made using an Applied Photophysics SX20 Stopped-flow Reaction Analyzer. Fluorescence was detected by exciting at 488 nm with 2 nm bandwidth and collecting emitted light at 520 nm through a 10-nm path at room temperature. The dead time of the instrument is 1.1 ms. For $k_{off}$, 2 μM of purified protein sample saturated with 200 mM L-lactate and 1 mM $CaCl_2$ was dissociated 1:1 with 100 mM EGTA at room temperature. Graphpad Prism was used to fit a single exponential dissociation for $k_{off}$. For eLACCO1.1, $k_{off}$ was faster than the dead time of the instrument, so a baseline fluorescence in the saturated state was obtained as a negative control. All measurements were done in triplicates, and error ± represents the s.e.m.

To collect the two-photon absorption spectra, the tunable femtosecond laser InSight DeepSee (Spectra-Physics, Santa Clara, CA) was used to excite the fluorescence of the sample contained within a PC1 Spectrofluorometer (ISS, Champaign, IL). The laser was automatically stepped to each wavelength over the spectral range with a custom LabVIEW program (National Instruments, Austin, TX), with 42 s at each wavelength to stabilize[30]. Two samples per laser scan were measured by using both the sample and reference holders and switching between them with the auto-switching mechanism on the PC1. The laser was focused on the sample through a 45-mm NIR achromatic lens, antireflection coating 750–1550 nm (Edmund Optics, Barrington, NJ). Fluorescence was collected from the first 0.7 mm of the sample at 90° with the standard PC1 collection optics through both 633/SP and 745/SP filters (Semrock, Rochester, NY) to remove all laser scattered light. To correct wavelength-to-wavelength variations of laser parameters, LDS798 (Exciton, Lockbourne, OH) in 1:2 $CHCl_3$:$CDCl_3$ was used as a reference standard between 912 and 1240 nm (ref. [31]), and coumarin 540 A (Exciton, Lockbourne, OH) in 1:10 DMSO:deuterated DMSO was used between 700 and 912 nm (ref. [32]). Adding the deuterated solvents (Millipore Sigma, Darmstadt, Germany) was necessary to decrease near-infrared solvent absorption. All the dye solutions were magnetically stirred throughout the measurements. Quadratic power dependence of fluorescence intensity in the proteins and standards was checked at several wavelengths across the spectrum.

For the L-lactate-bound state of eLACCO1.1, the two-photon cross section ($\sigma_2$) of the anionic form of the chromophore was measured versus rhodamine 6 G in MeOH at 976 and 960 nm ($\sigma_2$ (976 nm) = 12.7 GM, $\sigma_2$ (960 nm) = 10.9 GM)[33]. These $\sigma_2$ numbers closely agree with other literature data: ref. [34] at 976 nm (considering the correction discussed in ref. [33]), and ref. [35] at 960 nm. For the L-lactate-free state, the $\sigma_2$ of the neutral form of the chromophore was measured versus fluorescein (Millipore Sigma, Darmstadt, Germany) in 10 mM NaOH (pH 12) at 820 and 840 nm: $\sigma_2$ (820 nm) = 24.2 GM, $\sigma_2$ (840 nm) = 12.9 GM[33]. These $\sigma_2$ values for fluorescein also match other literature data[36,37]. Power dependence of fluorescence intensity was recorded with the PC1 monochromator at 550 nm (lactate bound) or 512 nm (lactate free) with the emission slits at a spectral width of 16 nm (full width at half maximum) and fitted to a parabola with the curvature coefficient proportional to $\sigma_2$. These coefficients were normalized for the concentration and the differential fluorescence quantum yield at the registration wavelength. The differential quantum yields of the standard and the sample were obtained with an integrating sphere spectrometer (Quantaurus-QY; Hamamatsu Photonics, Hamamatsu City, Japan) by selecting an integral range ~ 8 nm to the left and right of the registration wavelength. Concentrations were determined by Beer's Law. Extinction coefficients were determined by alkaline denaturation as detailed in ref. [38]. The two-photon absorption spectra were normalized to the $\sigma_2$ values at the two wavelengths and averaged. To normalize to the total two-photon brightness ($F_2$), the spectra were then multiplied by the quantum yield and the relative fraction of the respective form of the chromophore for which the $\sigma_2$ was measured. The data is presented this way because eLACCO1.1 contains a mixture of the neutral and anionic forms of the GFP chromophore. This is described in further detail in refs. [17,38].

**X-ray crystallography**. For crystallization, eLACCO1 cloned in pBAD-HisB with N-terminus 6×His tagged was expressed in *E. coli* DH10B strain. Briefly, a single colony from freshly transformed *E. coli* was inoculated into 500 mL of modified Terrific Broth (2% Luria-Bertani Broth supplemented with additional 1.4% tryptone, 0.7% yeast extract, 54 mM $K_2PO_4$, 16 mM $KH_2PO_4$, 0.8% glycerol, 0.2 mg mL$^{-1}$ ampicillin sodium salt) and incubated by shaking at 37 °C and 220 rpm for 8 h to reach exponential growth phase. Then, L-arabinose was added to 200 ppm to induce expression for another 48 h shaking at 30 °C and 250 rpm. Bacteria were then harvested and lysed using a sonicator (QSonica) for 4 cycles of 150 s sonication with 2 s off between each 1 s of sonication at 50 W power. After centrifugation at 13,000 g, the supernatant was purified with Ni-NTA affinity agarose bead (G-Biosciences) and eluted into PBS containing 250 mM imidazole. The eluted sample was further concentrated and desalted with an Amicon Ultra-15 Centrifugal Filter Device (Merck). The purified eLACCO1 protein was further applied on Superdex200pg (GE healthcare) size exclusion chromatography and the buffer was exchanged to TBS buffer supplemented with 1 mM $CaCl_2$. The fractions containing the eLACCO1 protein were pooled and concentrated to ~20 mg mL$^{-1}$, and then incubated with 2 mM L-lactate for the crystallization trail. Initial crystallization of the eLACCO1 protein was set up using 384-well plate via sitting drop vapor diffusion against commercially available kits at room temperature. The eLACCO1 crystal used for the data collection was grown in 0.2 M ammonium citrate dibasic and 20% w/v PEG 3350 by mixing 0.6 μL of reservoir solution with 0.6 μL of protein sample. The eLACCO1 crystals grew for 3–5 days and were cryoprotected with reservoir supplemented with 25% glycerol, and then frozen in liquid nitrogen. X-ray diffraction data were collected at 100 K at the Advanced Photon Source beamline line 23ID. The X-ray diffraction data were processed and scaled with XDS[39]. Data collection details and statistics were summarized in Table S1. The crystal structure of eLACCO1 was solved by molecular replacement approach implemented in Phaser program embedded in Phenix program package[40], using the GFP structure (PDB 3SG6) and the lactate binding protein (PDB 2ZZV) as search models[12,41]. One molecules of the eLACCO1 protein were present in the asymmetric unit. Further tracing of the missing residues and the structure were iteratively rebuilt in COOT and refined with Phenix program package[42,43]. The final model included the most of the residues except the N-terminal affinity tag and the glycine-rich linker to connect the original N- and C- termini of GFP. The final eLACCO1 structure bound with one molecule of L-lactate and one $Ca^{2+}$ ion was determined to 2.25 Å and refined to a final $R_{work}$/$R_{free}$ of 0.1484/0.1871 with high quality of stereochemistry. By generating symmetry mates, the eLACCO1 packed as a dimer in the crystal packing and shared a similar dimerization interface with TTHA0766 lactate binding protein (PDB 2ZZV).

**Construction of mammalian expression vectors**. For cell surface expression, the genes encoding eLACCO1.1, deLACCO, cpGFP, EGFP, and Laconic[6] (a gift from Luis Felipe Barros, Addgene plasmid #44238; http://n2t.net/addgene:44238; RRI-D:Addgene_44238) were amplified by PCR followed by digestion with BglII and EcoRI, and then ligated into pAEMXT vector (Covalys) that contains N-terminal leader sequence and C-terminal GPI anchor from CD59. To construct PDGFR-anchored eLACCO1.1 with various leader sequences, the gene encoding eLACCO1.1 including the CD59 leader and anchor sequence in the pAEMXT vector was first amplified by PCR followed by digestion with XhoI and HindIII, and then ligated into pcDNA3.1 vector (Thermo Fisher Scientific). Next, the gene encoding PDGFR transmembrane domain was amplified by PCR using pDisplay vector (Thermo Fisher Scientific) as a template, and then substituted with CD59 GPI domain of the pcDNA3.1 product above by using EcoRI and HindIII. Complementary oligonucleotides (Thermo Fisher Scientific) encoding each leader sequence were digested by XhoI and BglII, and then ligated into a similarly-digested pcDNA3.1 including PDGFR-anchored eLACCO1.1. The gene encoding mCherry was amplified by PCR followed by digestion with BglII and SalI, and then ligated into pDisplay vector that contains N-terminal Igκ leader sequence and C-terminal PDGFR transmembrane domain. To construct eLACCO1.1 plasmid for neural expression, the gene encoding eLACCO1.1 including the CD59 leader and anchor sequence in the pAEMXT vector was first amplified by PCR followed by digestion with NheI and XhoI, and then ligated into a human synapsin promoter vector (a gift from Jonathan Marvin). The gene encoding eLACCO1.1 was sub-cloned by restriction digests into pAAV plasmid containing the GFAP promoter (a gift from Michael Brenner) for AAV production.

**Imaging of eLACCO1.1 in HeLa, HEK293, and T98G cell lines**. HeLa and HEK293FT cells were maintained in Dulbecco's modified Eagle medium (Nakalai Tesque) supplemented with 10% fetal bovine serum (FBS; Sigma-Aldrich) and 1% penicillin-streptomycin (Nakalai Tesque) at 37 °C and 5% $CO_2$. T98G cells were maintained in minimum essential medium (Nakalai Tesque) supplemented with 10% FBS, 1% penicillin-streptomycin, 1% non-essential amino acid (Nakalai Tesque) and 1 mM sodium pyruvate (Nakalai Tesque) at 37 °C and 5% $CO_2$. Cells were seeded in 35-mm glass-bottom cell-culture dishes (Iwaki) and transiently transfected with the constructed plasmid using polyethyleneimine (Polysciences). Transfected cells were imaged using a IX83 wide-field fluorescence microscopy (Olympus) equipped with a pE-300 LED light source (CoolLED), a ×40 objective lens (numerical aperture (NA) = 1.3; oil), an ImagEM X2 EM-CCD camera (Hamamatsu), Cellsens software (Olympus) and a STR stage incubator (Tokai Hit). The filter sets used in live cell imaging had the following specification. eLACCO1.1, deLACCO, cpGFP, and EGFP: excitation 470/20 nm, dichroic mirror 490-nm dclp, and emission 518/45 nm; eLACCO1.1 (ratiometric imaging): excitation 365/10 nm, dichroic mirror 410-nm dclp, and emission 518/45 nm; mCherry: excitation 545/

20 nm, dichroic mirror 565-nm dclp, and emission 598/55 nm; Laconic (CFP): excitation 438/24 nm, dichroic mirror 458-nm dclp, and emission 483/32 nm; Laconic (FRET): excitation 438/24 nm, dichroic mirror 458-nm dclp, and emission 542/27 nm. Fluorescence images were analyzed with ImageJ software (National Institutes of Health).

For imaging of lactate-dependent fluorescence, HeLa cells transfected with eLACCO1.1, deLACCO, or eLaconic were washed twice with Hank's balanced salt solution (HBSS; Nakalai Tesque), and then 2 mL of HBSS supplemented with 10 mM HEPES (Nakalai Tesque) and 10 mM 2-deoxyglucose (Wako) was added to start the imaging at 37 °C followed by L-lactate stimulation. For photostability test, HeLa cells transfected with pAEMXT-eLACCO1.1, EGFP, or cpGFP were illuminated by excitation light at 100% intensity of LED (~10 mW cm$^{-2}$ on the objective lens) and their fluorescence images were recorded at 37 °C for 5 min with the exposure time of 50 ms and no interval time. Fluorescence intensities on cell membrane were collected after background subtraction. For imaging of L-lactate release, T98G cells transfected with eLACCO1.1 or deLACCO were washed twice with HBSS, and then 1.5 mL of HBSS supplemented with 10 mM HEPES and 1 mg mL$^{-1}$ (5.6 mM) or 4.5 mg mL$^{-1}$ (25 mM) D-glucose was added to start the imaging in the presence of DMSO (Wako), 1 μM AR-C155858 (Wako), 100 μM phloretin (Tokyo Chemical Industry), or 1 μM NCI-737 (a gift from Leonard Neckers). The fluorescence images were recorded for 2 h with the interval time of 15 min at 37 °C with time zero defined as the time point immediately preceding medium addition.

For imaging of Ca$^{2+}$-dependent fluorescence, HeLa cells seeded onto coverslips were transfected with eLACCO1.1. Forty-eight hours after transfection, the coverslips were transferred into Attofluor™ Cell Chamber (Thermo Fisher Scientific, Cat. #A7816) with modified Locke buffer (154 mM NaCl, 5.6 mM KCl, 1 mM MgCl$_2$, 5.6 mM D-glucose) supplemented with 20 mM HEPES and 10 mM L-lactate. Other bath solutions were supplemented with Ca$^{2+}$ of 0, 0.05, 0.1, 0.5, 1, and 2 mM. Rapid change of bath solutions during the image was performed in a remove-and-add manner using a homemade solution remover[44].

For in situ pH titration, HeLa cells seeded onto coverslips were co-transfected with pDisplay-pHuji (Addgene plasmid #61556) and pAMEXT-eLACCO1.1 or deLACCO. Forty-eight hours after transfection, the coverslips were transferred into Attofluor™ Cell Chamber with HBSS supplemented with 20 mM HEPES (Gibco, Cat. #15630130) and 10 mM 2-deoxy-D-glucose (Sigma-Aldrich Cat. #D8375-1G) at pH 7.05. Other bath solutions were supplemented with 10 mM L-lactate (Sigma-Aldrich Cat. #71718-10 G) and subsequently adjusted to their respective pH values. Rapid change of bath solutions during the image was performed in a remove-and-add manner using a homemade solution remover[44]. Cells are imaged on a Nikon Eclipse Ti-E epifluorescence microscope equipped with a 488 nm argon laser and a 543 nm He-Ne laser focused on the back aperture of a × 60 oil total internal reflection fluorescence (TIRF) objective lens (NA 1.49, Nikon). TIRF setup was achieved by a TI-TIRF-E Motorized Illuminator Unit (Nikon) to reduce the contribution of fluorophores that are not localized on the plasma membrane. Images were acquired every 10 s by a Photometrics QuantEM 512SC EM-CCD camera at a gain value of 500. To avoid the photoactivation artifacts, pHuji signal was acquired first in each cycle with the 543 nm laser with a TRITC filter cube followed by the acquisition of eLACCO1.1 signal with 488 nm laser and a FITC filter cube. NIS-Elements AR package software was used for automatic instrument control, data recording and measurement. Data was further analyzed and normalized to the intensity at pH 7.99 using a custom R script and plotted in GraphPad Prism software.

**Imaging of eLACCO1.1 in neurons.** Male and female P0 pups were obtained from a single timed-pregnant Sprague Dawley rat (Charles River Laboratories). Experiments were performed with rat cortical primary cultures from the P0 pups (pooled tissues from males and females), plated in glass-bottom 24-well plates where $0.5 × 10^6$ cells were used for three wells. Cultures were nucleofected at time of plating, and imaged 14 days later. Three wells were plated and imaged per nucleofected construct. Culture media was replaced with 1 mL of imaging buffer (145 mM NaCl, 2.5 mM KCl, 10 mM glucose, 10 mM HEPES, 2 mM CaCl$_2$, 1 mM MgCl$_2$, pH 7.4) for imaging[13]. Wide-field images were taken at the center of each well using a Nikon Eclipse microscope (20 × 0.4 NA, 488 nm excitation, 500–550 nm emission) at room temperature. These were the "APO" images. After a 20-min incubation in the presence of ~10 mM L-lactate, the same fields of view were recorded again. These were the "SAT" images. APO and SAT images were aligned by cross-correlation. A pixel classifier (Ilastik[45]) was trained using manual annotations to label each pixel in the images as background, neurite, soma, or intracellular inclusion. A scalar constant background was subtracted from all images to account for camera offset. Reported values of $\Delta F/F$ for pixels classified as neurites were calculated as (sum of neurite pixel intensities SAT)/(sum of neurite pixel intensities APO) − 1.

**Surgery and in vivo microinjections of adeno-associated virus (AAV).** AAV was packaged in HEK293 cells by triple transfection of pAAV plasmids encoding eLACCO1.1, a rep/cap encoding plasmid (pXR5 or POM2/9) and the helper plasmid pXX680. At 120 h after the transfection, culture medium was collected, filtered and concentrated through a 100 kDa TFF cassette. Viral particles were purified by ultracentrifugation on an iodixanol gradient, washed on a 100 kDa

MWCO spin column and stored in PBS containing 10% D-sorbitol and 0.002% F-68 pluronic acid. Viral titers (GC/mL) were determined by ddPCR using universal primers binding the ITRs. Transduction efficiency was validated in primary rat neuron and astrocyte cultures. Male mice (C57bl/6, P45) were anesthetized via isoflurane (5% for induction, 2–3% for maintenance, v/v). Depth of anesthesia was determined by observing breathing rates and toe-pinch ensured proper loss of reflexes. Following deep anesthesia, mice were head fixed on a stereotaxic apparatus (David Kopf Instruments) with a bite bar and ear bars, with ventilated anesthesia administration. The mice were injected with 0.05 μL of buprenorphine subcutaneously (Buprenex, 0.1 mg mL$^{-1}$), and artificial tears were applied to the eyes before beginning surgery. The hair on the scalp was removed prior to surgery, and the incision was washed with 10% povidone iodine and 70% ethanol, 3 times each, alternating. An incision was made on the scalp to expose bregma and the craniotomy site with coordinates are as follows. Somatosensory cortex: −1.58 mm posterior and +3.0 and −3.0 mm lateral (for bilateral injection) from bregma, and 0.7–0.5 mm ventrally from the pial surface. A 2–3 mm craniotomy was made at the injection site using a small burr (Fine Science Tools), powered by a drill (K.1070, Foredom). Saline (0.9%) was applied to keep the skull cool, to maintain skin hydration, and to remove bone debris. AAVs were injected via a beveled borosilicate pipette (World Precision Instruments) by a Nanoject 2 (Drummond Scientific). Five, 69 nL injections were given at each site, totaling 345 nL of virus was infused into each region of somatosensory cortex, and each virus contained the GFAP promoter driving the following constructs at the indicated titer: eLACCO1.1 ($1.5 × 10^{13}$ gC mL$^{-1}$), deLACCO ($1.5 × 10^{13}$ gC mL$^{-1}$). Following injection, the needle was left in place for 10 min to allow for fluid pressure normalization. Following needle withdraw, scalp was sutured with silk sutures and mice were closely monitored, kept on a heating pad and given buprenorphine twice daily for 48 h post-op (0.05 mL, 0.1 mg mL$^{-1}$), and fed chow with sulfonamide sulfadiazine trimethoprim (32 g kg$^{-1}$) for 1 week post-op.

**Preparation of acute cortical brain slice.** At 6 weeks after the AAV injection, mice were anaesthetized with gaseous isoflurane (5%) and then decapitated. The brain was removed, then submerged for 2 min in ice-cold slicing solution containing (in mM): 119.9 N-methyl-D-glucamine, 2.5 KCl, 25 NaHCO$_3$, 1.0 CaCl$_2$-2H$_2$O, 6.9 MgCl$_2$-6H$_2$O, 1.4 NaH$_2$PO$_4$-H$_2$O, and 20 D-glucose. The brain was then Krazy Glued onto a vibratome tray (Leica Instruments, VT1200S) and then re-submerged in ice-cold slicing solution. Acute coronal slices were prepared from the somatosensory cortex (400 μm thick) using a vibratome. The slices were incubated for 45 min at 33 °C in a recovery chamber filled with artificial cerebrospinal fluid containing (in mM): 126 NaCl, 2.5 KCl, 25 NaHCO$_3$, 1.3 CaCl$_2$-2H$_2$O, 1.2 MgCl$_2$-6H$_2$O, 1.25 NaH$_2$PO$_4$-H$_2$O, and 10 D-glucose. After recovery, slices were stained with astrocyte marker sulforhodamine 101 (SR101) to visualize features to maintain focal plane during imaging. Throughout, brain slices were continuously supplied with carbogen (95% oxygen, 5% CO$_2$).

**Two-photon microscopy of eLACCO1.1 in acute cortical brain slice.** Brain slices were imaged using a custom built two-photon microscope[46] fed by a Ti:Sapphire laser source (Coherent Ultra II, ~4 W average output at 800 nm, ~80 MHz). Image data were acquired using MatLab (2013) running the open source scanning microscope control software ScanImage (version 3.81, HHMI/Janelia Research Campus). Imaging was performed at an excitation wavelength of 940 nm. The microscope was equipped with a primary dichroic mirror at 695 nm and green and red fluorescence was split and filtered using a secondary dichroic at 560 nm and two bandpass emission filters: 525–40 nm and 605–70 nm (Chroma Technologies). Time series images were acquired at 0.98 Hz with a pixel density of 512 by 512 and a field of view size of ~292 μm. Imaging used a ×40 water dipping objective lens (NA 1.0, WD 2.5 mm, Zeiss). Imaging was performed at room temperature. Brain slices were superfused with L-lactate (Sigma Aldrich) at concentrations of 1, 2.5, and 10 mM. $\Delta F/F$ was calculated by: $\Delta F/F = ((F_x − F_b)/F_b)$, where $F_x$ is the peak fluorescence intensity and $F_b$ is the baseline fluorescence intensity. Regions of interest were selected based on identifying fine processes via SR-101 fluorescence that did not shift focal plane during the duration of imaging.

**Animal care.** For experiments performed at University of Calgary, all methods for animal care and use were approved by the University of Calgary Animal Care and Use Committee and were in accordance with the National Institutes of Health Guide for the Care and Use of Laboratory Animals. For experiments at HHMI Janelia Research Campus, all surgical and experimental procedures were in accordance with protocols approved by the HHMI Janelia Research Campus Institutional Animal Care and Use Committee and Institutional Biosafety Committee.

**Statistics and reproducibility.** All data are expressed as mean ± s.d. or mean ± s.e.m., as specified in figure legends. Box plots with and without notches are used for Figs. 4b, c, 3d, e, and 5d, respectively. In these plots, the horizontal line is the median; the top and bottom of the notch denote the 95% confidence interval of the median; the top and bottom horizontal lines are the 25th and 75th percentiles for the data; and the whiskers extend one standard deviation range from the mean represented as black filled circle. Sample sizes ($n$) are listed with each experiment.

No samples were excluded from analysis and all experiments were reproducible. In photobleaching experiments, group differences were analyzed using one-way ANOVA (Igor Pro 8). In eLACCO1.1 imaging (Fig. 3e) and brain slice experiments, the significant differences were analyzed using Student's $t$-test (Graphpad Prism). Microsoft Excel software was used to plot for Figs. 1b, c, 2a, b, d, and 6c, f.

**Reporting summary**. Further information on research design is available in the Nature Research Reporting Summary linked to this article.

## Data availability

Structure coordinates of eLACCO1 have been deposited in the Protein Data Bank with a code of 7E9Y. pBAD-eLACCO1.1 (plasmid no. 167944), pBAD-deLACCO (plasmid no. 167945), pAEMXT-eLACCO1.1 (plasmid no. 167946), and pAEMXT-deLACCO (plasmid no. 167947) are available via Addgene [https://www.addgene.org/browse/article/28216216/]. pAAV-hSyn-eLACCO1.1 (plasmid no. 168788), pAAV-CAG-eLACCO1.1 (plasmid no. 168789), and pAAV-GFAP-eLACCO1.1 (plasmid no. 168790) are also available via Addgene [https://www.addgene.org/browse/article/28216308/]. Source data are provided with this paper.

## Code availability

Custom code is available from the corresponding author, and at https://github.com/shucez/eLACCO_manuscript_TIRF_deltaF_F0.

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

## Acknowledgements

The authors thank the University of Alberta Molecular Biology Services Unit, Y. Shen, L. Zarowny, G. Nguyen, D. Qi and S. Hario for technical support. We thank H. Yoshimura for the CD59 gene. We thank L. Neckers for the LDH inhibitor NCI-737. Work at the University of Tokyo was supported by the Japan Society for the Promotion of Science (Grants-in-Aid for Early-Career Scientists 19K15691 and Grants-in-Aid for Scientific

Research S 19H05633), Toyota Physical and Chemical Research Institute, and The Precise Measurement Technology Promotion Foundation. Work at the University of Alberta was supported by Natural Sciences and Engineering Research Council of Canada (NSERC) grant RGPIN-2018-04364 and Canadian Institutes of Health Research (CIHR) grant FS-154310. Work at Montana State University was supported by National Institutes of Health (NIH) grants U01 NS094246, U24 NS109107, and F31 NS108593. Y.W was supported by the Alberta Parkinson Society Fellowship and National Natural Science Foundation of China (Nos. 31870132 and 82072237).

## Author contributions

Y.N. developed eLACCO1.1 and performed in vitro characterization and endogenous L-lactate imaging. R.S.M. and M.D. measured one-photon absorbance spectra and two-photon excitation spectra. C.M.R. and J.H. performed acute brain slice imaging. S.Z. and Y.W. worked on the crystallography of eLACCO1. S.Z. performed in situ pH titration. A.A. performed stopped-flow experiment. A.A. and K.P. performed the imaging of primary neurons and data analysis. Y.K. performed screening of leader sequence. M.-E.P produced AAV. M.J.L., K.P., J.S.B., G.R.G., and R.E.C. supervised research. Y.N. and R.E.C. wrote the manuscript.

## Competing interests

The authors declare no competing interests.
