## [Peer Review File · Nature Communications]

Reviewers' Comments:

Reviewer #1:

Remarks to the Author:

Nasu & Campbell and co-workers report the development of the first genetically encoded fluorescent sensor of extracellular lactate, eLACCO1.1. Although intracellular sensors of lactate have previously been developed, the authors demonstrated that at least one of these sensors was not functional when expressed on the cell surface, and therefore the novelty and need specifically for an extracellular lactate sensor was clearly motivated. They demonstrate it has very good dynamic range both in vitro and in brain slices in practice, excellent kinetics, a decent lactate affinity for a first-generation sensor, and it can be used effectively with either one-photon or two-photon fluorescence excitation. They also demonstrate that it has very good substrate specificity for lactate with the exception of the expected pH sensitivity for cpGFP-based sensors, and therefore they created a non-binding sensor deLACCO that can be used as a control for pH bias or other non-specific changes. Overall, the development of eLACCO1.1 could greatly enhance the study of metabolic coupling between cells, potentially even beyond the astrocyte-neuron lactate shuttle hypothesis that they discuss.

While the work is very exciting and the manuscript was an enjoyable read, the presentation of the results could be improved to make it more appropriate for publication for both the broader and specialist audiences. One major comment is that it is not clear why the authors compressed their results and discussion into such an overly compact manuscript. Too much of the relevant and important information that frames the work is relegated to the SI, and it is not clear why all the data that was included in the main text was compressed into just 2 figures. The lack of discussion and clearly readable data figures should be addressed to make the work accessible. Specific comments include:

- The T98G glioma cell experiments were initiated by the addition of a high glucose concentration of 25 mM. It would be potentially be more convincing that eLACCO1.1 would be useful for studying intercellular metabolic coupling if a direct intervention on intracellular metabolic pathways were studied rather than just blockade of lactate export. For example, could eLACCO1.1 detect something like inhibition of glycolytic production of pyruvate, which fuels lactate production, with a GAPDH inhibitor such as iodoacetamide.
- Figure 1 should be split into 2 if not 3 or more figures at the very least for readable presentation. For example, the different series in 1d and 1e are not easily distinguishable at such a small size, and it seems unnecessary to shrink these important data when the manuscript seems to be well below the article limits for NCOMM.
- The photophysical parameters reported in Supp Table 2, which represents key characterization of the sensor, seems like it should be in the main text also.
- Page 2 Line 47 The authors should provide at least some rationale for the choice of their lactate binding domain, and perhaps later in the text at least discuss briefly the optimization they did in SI Fig. 2 for the homologues. What are the advantages to using TTHA0766 versus LdR used in Laconic and Lindoblum? They could also give more rational for choice of the solvent exposed loops as cpGFP insertion sites shown in SI Fig 1.
- It could be beneficial to move SI Fig 3 to the main text, especially as it visually illustrates the engineering process and that in the end mutations both in the cpGFP and TTHA0766 modules were necessary for a functional eLACCO1.
- Page 3 Line 61-63: All the discussion of the X-ray crystal structure and sensing mechanism and structural basis of the calcium requirement are in the SI, and it would be incredibly beneficial to have this moved to the main text. Related to this, the authors discuss in SI Fig 6 that attempts to monomerize eLACCO1 caused loss of function – how does dimerization affect the proposed sensing mechanism in SI Note 1 and related affinity tuning in SI Fig 7. What is the dimer affinity? Is there likely to be an equilibrium of both dark monomeric and fluorescent dimeric sensor when expressed?

- Page 4 Line 87: Given that deLACCO is proposed as a control, the pH sensitivity of eLACCO1.1 and deLACCO should be shown to be comparable or at least discussed if, as the authors state, it originates entirely from the cpGFP module.

- Starting at Page 4 Line 88: Again, given that optimization of cell surface display is a major part of the development of this extracellular sensor, it would be greatly beneficial to move perhaps some combined version of SI Fig 9 and SI Fig 10 to the main text.

- Page 4 Lines 100-102: The panels in SI Fig 8 appear to be mislabeled (b and d switched?)

- Why is the expression so punctate on culture neurons?

- Regarding the slice experiment in Fig 2a-d, given how other sensors operate in slice or in vivo it is not too surprising that some attenuation to the dynamic range is observed. However, it would be helpful to have the authors perspective on what might cause this. In particular, have experiments been done in which slices have been treated with LDH or other lactate degrading enzyme to determine if basal lactate concentration is high in these acute slice preps, which could cause the attenuation in dynamic range?

Overall, the manuscript was convincing. With the exception of the first comment which could merit a just a few additional experiments, the manuscript could be made appropriate for publication be several major edits to the organization itself. Specifically, to do justice to the work much of the data in the SI should be reincorporated into the main text and appropriately discussed.

Reviewer #2:

Remarks to the Author:

The manuscript presents a new green fluorescent genetically encoded biosensor, eLACCO1.1, that can be targeted to the surface of mammalian cells and used to report changes in extracellular [L-lactate]. It is the first reported sensor of extracellular lactate, and due to the authors' substantial effort at in vitro directed evolution, it displays large intensity changes in response to lactate. The sensor is specific for lactate over other chemically similar compounds, but it has a requirement for calcium and therefore cannot be used to measure intracellular [lactate].

The authors propose that this sensor should be of use in investigations of cell-to-cell shuttles of L-lactate, such as the astrocyte-neuron lactate shuttle. However, the value of the extracellular sensor for such studies is not completely clear, as a detected change in extracellular [lactate] could not easily help determine the cellular source of lactate (e.g. neurons or astrocytes?) or the directionality of lactate movement between cells. It could certainly be valuable to learn about the changes in extracellular [lactate] in different brain regions under different conditions (as previously done with microdialysis). Both for investigating the shuttle and for a generic characterization of lactate behavior, it would be important to establish that the sensor can be used for quantitative measurement of [lactate] and not just visualization of changes. In principle, this should be possible, as the sensor is ratiometric with different excitation wavelengths; but ratio imaging to obtain calibrated estimates of [lactate] was not done in any of the trial experiments here.

In general, the trial experiments here do not help much to assess the usefulness of the new probe for monitoring physiological changes in lactate. The brain slice experiments simply measure the response of the sensor to bath-applied lactate, rather than to endogenous production and release of lactate. The sizes of these responses are also very variable (Fig. 2d). The glioma responses to altered [glucose] show some change, presumably in endogenously produced lactate, but the manipulation is very large (0 to 25 mM glucose) and the response is quite slow. Also, although the sensor is likely to be fully extracellular, no experiments are done to test this (for instance, insensitivity of the response to blockade of the transporter).

The final version of the sensor with an affinity suitable for physiologically-relevant measurements (eLACCO1.1) is missing some important characterizations. Most importantly, its responsiveness

to/requirement for Ca²⁺ is not presented. While the prototype sensor, eLACCO1, is reported to require near micromolar Ca²⁺ for lactate responses, the same information is not provided for the mature sensor, eLACCO1.1. The crystal structure shows that calcium is bound in concert with lactate in the binding pocket, so given that the lactate affinity was reduced by 1,000-fold by making a mutation in the binding pocket, one cannot assume that Ca²⁺ affinity remains constant. Given that the two ligands bind together, the [Ca²⁺] may also affect the dose-response midpoint for lactate. Likewise, the authors only present binding specificity data (Sup. Fig. 5c) and pH responses (Sup. Fig. 5d) for eLACCO1, but not eLACCO1.1.

Minor

1. The legend of Fig. 2e does not adequately describe the experiment being performed; there is no mention that glucose is being applied, and it is presumably the glucose that is driving lactate production. In addition to inhibiting the MCT, phloretin is known to inhibit glucose transporters. A more specific MCT inhibitor (e.g. AR-C155858) would probably have been better.
2. In Fig 2d, the variability is enormous, which the authors do not comment on. Given this variability, the "representative traces" shown in Fig. 2c are not sufficient (need to show mean +/- standard error) nor are they representative (the trace in 2c shows a >1.5-fold change in $\Delta F/F$ in response to 10 mM lactate, while the y-axis in 2d only goes to 1.2).
3. Main text, line 61: The sentence describing the specificity for lactate refers to Sup Fig. 5d, but it should refer to Sup Fig. 5c.
4. The photobleaching and dose-response panels of Supp. Fig. 8 (b and d) have been reversed.

Reviewer #3:

Remarks to the Author:

This is a very interesting manuscript in which Nasu and colleagues designed and developed a genetically-encoded fluorescent biosensor that detects extracellular L-lactate. The manuscript is nicely described and elegantly presented, and the experimental work behind the development of this biosensor is very robust, leading to the proposal of a biosensor likely suitable for the assessment of intercellular exchange of L-lactate between, e.g., astrocytes and neurons -although it might be useful for determining L-lactate exchange amongst cancer cells, for instance. Therefore, it seems reasonable that the many researchers using this biosensor should know in more detail to which extension its sensitivity and/or selectivity is/are influenced by pH and calcium, two important factors that enormously change during neurotransmission. Therefore, these issues should be reinforced according to the following comments.

Comments

1. As the authors are aware, one of the major problems with protein-based biosensors is their vulnerability to protonation making them highly sensitive to pH changes. This is particularly important during physiological neurotransmission, when brain pH may shift between 6.5 and 8.0 [M Chesler & K Kaila (1992) Modulation of pH by neuronal activity. Trends Neurosci 10:396-402. doi: 10.1016/0166-2236(92)90191-a. PMID: 1279865]. The authors indeed aimed to address this issue by determining the normalized fluorescence intensity of eLACCO1 with or without 10 mM L-lactate in vitro (i.e., in the absence of cells) in a wide range of pH (Supplementary Fig. 5d). According to Suppl. Fig. 5d, the largest change in normalized fluorescence intensity took place up to pH value of 7.0 (with L-lactate). However, the experimental values were mathematically adjusted to a sigmoid curve leading to the apparent conclusion that the normalized fluorescence intensity did not change in a pH range between 7 and 10. However, looking at the actual data points, the normalized fluorescence intensity progressively increased as from pH 6.5, i.e. a pH value that is within the physiological pH range during neurotransmission (Chesler & Kaila, 1992). Admittedly, the observed changes in normalized fluorescence intensity between pH values of 7 and 10 are subtle -despite progressive suggesting an effect; however, and importantly, in this experiment the fluorescence intensity values were normalized to that obtained at pH 2, hence abrogating the possibility to ascertain how much is the actual change size in absolute fluorescence, particularly in the physiological pH range 6.5-8. Given the critical importance of this issue, especially for the neuroscience community who, according to the authors, will likely be interested

in this biosensor, it would be needed (i) to represent the graph of Suppl. Fig. 5d using the raw data (i.e., not the normalized fluorescence), and (ii) given the likely possibility that the biosensor's pH sensitivity may be different in a physiological setting, the authors should also perform a simple experiment that shows the unnormalized fluorescence intensity values in cells expressing eLACCO1 at a fixed L-lactate concentration (e.g., the K_d) in the pH range around that physiologically achievable (i.e., from 6.5 to 8).

2. Similarly to the previous point, calcium (as magnesium) is an important messenger during neurotransmission and, accordingly, the authors addressed how much the eLACCO1 normalized fluorescence intensity changed with calcium (and magnesium) concentrations. Unfortunately, the normalized data likely avoids the observation of possible subtle changes in fluorescence intensities at calcium concentrations at the synaptic cleft during neurotransmission, which varies from 0 to 2 mM [Cohen JE, Fields RD. (2004) Extracellular calcium depletion in synaptic transmission. *Neuroscientist* 10(1):12-7. doi: 10.1177/1073858403259440. PMID: 14987443]. Moreover, during neurotransmission, a 0 mM calcium concentration is easily achievable, hence it remains unknown whether the L-lactate detection by the eLACCO1 biosensor would be affected. This is also important for cancer research, given the extreme changes in extracellular calcium concentration that can be very high or 0. Therefore, the authors should perform a simple experiment to show the unnormalized fluorescence intensity values in cells expressing eLACCO1 at a fixed L-lactate concentration (e.g., the K_d) in the extracellular calcium range around that achievable physiologically (i.e., from 0 to 2 mM).

Dear Dr. Eldridge and all reviewers,

We greatly appreciate the insightful and constructive comments from all the reviewers. These comments have proven very helpful for improving the manuscript. To address the comments, we have revised our manuscript in a point-to-point manner as described on the following pages. The table below summarizes the revised numbering of figures and tables. Newly added figures are shown in blue.

New	Old	Title
Main figures		
Fig. 1	Fig. S3a-c Fig. 1a,b	Development of a genetically encoded L-lactate biosensor, eLACCO1.
Fig. 2	Fig. 1c-e Fig. S8a	In vitro characterization of affinity-tuned eLACCO1.1. (+ new Fig. 2c,e)
Fig. 3	Fig. 1f,g Fig. S10	Targeting of eLACCO1.1 to the extracellular environment. (+ new Fig. 3e)
Fig. 4	Fig. 1h-j Fig. 2a-d Fig. S8b-d	Characterization of eLACCO1.1 in live mammalian cells and acute brain slice.
Fig. 5	Fig. 2e,f	Imaging of endogenous L-lactate release from glioblastoma cells. (+ new Fig. 5a, b (AR-C155858), c (AR-C155858) and d-f)
Supplementary figures		
Fig. S1	Fig. S1	Construction of the biosensor prototype.
Fig. S2	Fig. S2	Biosensor prototypes based on the various TTHA0766 homologues.
Fig. S3	Fig. S4	Sequence alignment of TTHA0766, cpGFP, and eLACCO1.
Fig. S4	Fig. S5	In vitro characterization of eLACCO1.
Fig. S5	Fig. S6	Crystal structure of eLACCO1. (+ new Fig. S5b)
Fig. S6	Fig. S7	Affinity tuning of eLACCO1.
Fig. S7		In vitro characterization of deLACCO.
Fig. S8	Fig. S9	Membrane trafficking of eLACCO1.1 with various leader sequences.
Fig. S9		Ratiometric imaging of eLACCO1.1 on live HeLa cells.
Fig. S10		Ca ²⁺ titration on live HeLa cells.
Fig. S11		pH titration on live HeLa cells.
Fig. S12	Fig. S11	Stopped-flow analysis of eLACCO1 and eLACCO1.1.
Fig. S13	Fig. S12	Attempted imaging of extracellular L-lactate with Laconic.
Tables		
Table 1	Table S2	One- and two-photon photophysical parameters of eLACCO1.1.
Table S1	Table S1	Crystallographic and refinement statistics of eLACCO1.

Reviewer #1

Nasu & Campbell and co-workers report the development of the first genetically encoded fluorescent sensor of extracellular lactate, eLACCO1.1. Although intracellular sensors of lactate have previously been developed, the authors demonstrated that at least one of these sensors was not functional when expressed on the cell surface, and therefore the novelty and need specifically for an extracellular lactate sensor was clearly motivated. They demonstrate it has very good dynamic range both in vitro and in brain slices in practice, excellent kinetics, a decent lactate affinity for a first-generation sensor, and it can be used effectively with either one-photon or two-photon fluorescence excitation. They also demonstrate that it has very good substrate specificity for lactate with the exception of the expected pH sensitivity for cpGFP-based sensors, and therefore they created a non-binding sensor deLACCO that can be used as a control for pH bias or other non-specific changes. Overall, the development of eLACCO1.1 could greatly enhance the study of metabolic coupling between cells, potentially even beyond the astrocyte-neuron lactate shuttle hypothesis that they discuss.

Thank you for your positive comments and your enthusiasm for this work.

While the work is very exciting and the manuscript was an enjoyable read, the presentation of the results could be improved to make it more appropriate for publication for both the broader and specialist audiences. One major comment is that it is not clear why the authors compressed their results and discussion into such an overly compact manuscript. Too much of the relevant and important information that frames the work is relegated to the SI, and it is not clear why all the data that was included in the main text was compressed into just 2 figures. The lack of discussion and clearly readable data figures should be addressed to make the work accessible.

Thank you for your supportive and helpful comments. The manuscript was originally formatted as a brief communication for a journal with a lower word count limit. We have now expanded the representation and discussion of the results to make the work more accessible.

- The T98G glioma cell experiments were initiated by the addition of a high glucose concentration of 25 mM. It would be potentially be more convincing that eLACCO1.1 would be useful for studying intercellular metabolic coupling if a direct intervention on intracellular metabolic pathways were studied rather than just blockade of lactate export. For example, could eLACCO1.1 detect something like inhibition of glycolytic production of pyruvate, which fuels lactate production, with a GAPDH inhibitor such as iodoacetamide.

Thank you for your suggestion. In an attempt to perform the suggested experiment, we stimulated eLACCO1.1-expressing T98G cells with 100 \$\mu\$ M iodoacetate, an inhibitor of GAPDH, in the presence of a physiologically relevant concentration of glucose (5.6 mM). Unexpectedly, this stimulation resulted in apparent aggregation of eLACCO1.1 (please see the figure attached below). Iodoacetate modifies cysteine residues via alkylation. eLACCO1.1 has cysteine residues at position of 340 and 362 (Supplementary Figure 3) and we suspect that iodoacetate might induce aggregation due to the modification of these cysteine residues.

To avoid this iodoacetate-induced aggregation artefact, we turned to using NCI-737, an inhibitor of lactate dehydrogenase (LDH). Inhibition of LDH should prevent conversion of pyruvate to lactate and would be expected to lead to a decrease in lactate exported from the cell. Indeed, this is what we observed, as shown in Figure 5d-f and described in the following sentences in the main text (lines 195–202), *“To image the production and export of endogenous L-lactate under a more physiologically relevant condition, we observed eLACCO1.1 expressing T98G cells treated with a physiological plasma concentration of glucose (5.6 mM, Fig. 5d-f)²⁰. Treatment with NCI-737, an inhibitor of lactate dehydrogenase (LDH), caused a slight increase in the fluorescence response of deLACCO. In contrast, eLACCO1.1 showed a decrease in the fluorescence response upon NCI-737 treatment. The opposite effects of NCI-737 on the responses of eLACCO1.1 and deLACCO are consistent with the observed fluorescence changes being due to changes in the extracellular L-lactate concentration.”*

- Figure 1 should be split into 2 if not 3 or more figures at the very least for readable presentation. For example, the different series in 1d and 1e are not easily distinguishable at such a small size, and it seems unnecessary to shrink these important data when the manuscript seems to be well below the article limits for NCOMM.

To address this comment, we have split the original Figure 1 into four figures (Figures 1–4). In this process, we moved the *in vitro* and live cell characterization of eLACCO1.1 (the original Supplementary Figure 8) to Figures 2 and 4.

- The photophysical parameters reported in Supp Table 2, which represents key characterization of the sensor, seems like it should be in the main text also.

To address this comment, we have moved the original Supplementary Table 2 (photophysical parameters of eLACCO1.1) to the main text as Table 1.

- Page 2 Line 47 The authors should provide at least some rationale for the choice of their lactate binding domain, and perhaps later in the text at least discuss briefly the optimization they did in SI Fig. 2 for the homologues. What are the advantages to using TTHA0766 versus LldR used in Laconic and

Lindoblum? They could also give more rationale for choice of the solvent exposed loops as cpGFP insertion sites shown in SI Fig 1.

To better clarify the rationale for the choice of the TTHA0766 protein, we have now added the following text to the main text (lines 52–54), *“Periplasmic binding proteins derived from prokaryotic organisms have proven to be particularly effective sensing domain for extracellular single fluorescent protein-based biosensors⁵, with key examples that include glutamate⁸, GABA⁹, acetylcholine¹⁰ and serotonin¹¹.”*

To more clearly explain the construction of biosensor prototype using the TTHA0766 homologues, we have now moved the following sentences from the Supplementary Figure 2 to the main text (lines 62–65), *“Efforts to create prototype biosensors by inserting cpGFP into the analogous regions of TTHA0766 homologues produced no variants with larger L-lactate-dependent changes in fluorescence intensity (**Supplementary Fig. 2**). Accordingly, we focused our efforts on further development of eLACCO0.1.”*

To more clearly explain the potential advantage of TTHA0766 over LldR, we have now added the following paragraphs to the Discussion section of the main text (lines 216–238):

*The choice of sensing domain is critically important to the development of any genetically encoded biosensor⁵. Biosensors for extracellular applications (e.g., those with specificity for neurotransmitters like glutamate⁸, GABA⁹, acetylcholine^{10,21}, serotonin^{11,22,23}, dopamine^{24,25} and norepinephrine²⁶) have generally used microbial periplasmic binding proteins (PBPs) or G-protein-coupled receptors (GPCRs) as the sensing domain. In this work, we chose to use *Thermus thermophilis* TTHA0766 L-lactate-binding PBP. Relative to other possible sensing domains that are normally found in the reducing environment of the cytoplasm, an advantage of microbial PBPs is that they naturally function in the oxidative environment of the periplasm. This property is likely beneficial to PBP-based biosensors for extracellular targets, since the protein must be trafficked through oxidative organelles such as the endoplasmic reticulum and Golgi apparatus and ultimately exposed to the oxidative extracellular environment. A gluconate operon repressor family transcription factor, LldR, has previously been used as the sensing domain for intracellular L-lactate biosensors^{6,7}. Attempts to use the Laconic LldR-based biosensor for imaging of extracellular L-lactate were unsuccessful (**Fig. 4b** and **Supplementary Fig. 13**), consistent with the suggested importance of using sensing domains that are compatible with the extracellular environment.*

Relative to GPCR-based biosensors, there are several technical advantages associated with development of PBP-based single FP-based biosensors. Of particular relevance to the methods used in this work, directed evolution with bacterial expression is applicable to PBP-based biosensors which can be expressed in soluble form, but is not applicable to integral membrane proteins such as GPCRs. Furthermore, these soluble biosensors can often be crystallized in order to resolve their structure at atomic resolution, providing insights that guide and accelerate engineering efforts. These insights ultimately help to produce biosensors with fluorescent responses that are substantially larger than those yet demonstrated for GPCR-based biosensors.

To better describe the rationale for choice of sites for inserting cpGFP into TTHA0766, we have now moved the following sentence from the Supplementary Figure 1 to the main text (lines 57–59), *“Positions of insertion sites in TTHA0766 were chosen by manual inspection of the protein crystal structure to identify loop regions that were solvent exposed and likely to undergo L-lactate dependent conformational changes^{5,12}.”*

- It could be beneficial to move SI Fig 3 to the main text, especially as it visually illustrates the engineering process and that in the end mutations both in the cpGFP and TTHA0766 modules were necessary for a functional eLACCO1.

To address this comment, we have moved the original Supplementary Figure 3 to the main Figure 1. Mutations in eLACCO1 relative to cpGFP and TTHA0766 are shown in Supplementary Figure 3 (the original Supplementary Figure 4).

- Page 3 Line 61-63: All the discussion of the X-ray crystal structure and sensing mechanism and structural basis of the calcium requirement are in the SI, and it would be incredibly beneficial to have this moved to the main text. Related to this, the authors discuss in SI Fig 6 that attempts to monomerize eLACCO1 caused loss of function – how does dimerization affect the proposed sensing mechanism in SI Note 1 and related affinity tuning in SI Fig 7. What is the dimer affinity? Is there likely to be an equilibrium of both dark monomeric and fluorescent dimeric sensor when expressed?

Thank you very much for comments on the structure and sensing mechanism of eLACCO1. To make the discussion of the mechanism more accessible, we have now moved the original Supplementary Note 1 (description of the X-ray crystal structure) to the main text (lines 78–88) and Discussion (lines 239–258).

The Hill coefficient of eLACCO1 is close to unity, suggesting that the protomers in the dimer do not interact cooperatively (Supplementary Fig. 4c). This non-cooperative target binding is consistent with a previously-reported dimeric protein TakP which is a tripartite ATP-independent periplasmic protein similar to TTHA0766 (Gonin, S. et al. *BMC Structural Biology* 7, 11 (2007)). The dimerization of eLACCO1 may be of critical importance to the protein folding but irrelevant to L-lactate sensing.

To estimate the affinity of eLACCO1-eLACCO1 interaction, we analyzed a size exclusion chromatograph of eLACCO1 for the crystallography and have included the data as Supplementary Figure 5b. The analysis resulted in an estimated dissociation constant (K_d) of 66 nM.

- Page 4 Line 87: Given that deLACCO is proposed as a control, the pH sensitivity of eLACCO1.1 and deLACCO should be shown to be comparable or at least discussed if, as the authors state, it originates entirely from the cpGFP module.

To better characterize the pH sensitivity of affinity-tuned eLACCO1.1 and control biosensor deLACCO, we have now added the pH titration data in the main Figure 2e (eLACCO1.1) and Supplementary Figure 7 (deLACCO), and the following sentences to the main text (lines 116–119), “*The fluorescence of eLACCO1.1 is pH dependent, exhibiting pK_a values of 7.4 and 9.4 in the presence and absence of L-lactate, respectively (Fig. 2e). The control biosensor deLACCO showed no response to L-lactate, and pH dependence that was similar to the lactate-free state of eLACCO1.1 (Supplementary Fig. 7).*”

- Starting at Page 4 Line 88: Again, given that optimization of cell surface display is a major part of the development of this extracellular sensor, it would be greatly beneficial to move perhaps some combined version of SI Fig 9 and SI Fig 10 to the main text.

We have moved the original Supplementary Figure 10 to the main Figure 3.

- Page 4 Lines 100-102: The panels in SI Fig 8 appear to be mislabeled (b and d switched?)

Thank you for pointing out. We moved the original Supplementary Figure 8b-d (the live cell characterization of eLACCO1.1) to Figure 4 and this issue has now been resolved.

- Why is the expression so punctate on culture neurons?

We had noticed the punctate expression but, as of yet, have no specific insight as to the mechanistic reason why eLACCO1.1 shows punctate fluorescence on primary culture neurons. A similar phenomenon has been reported for the extracellular glutamate biosensor iGluSnFR (Marvin, J. et al. *Nature Methods* **10**, 162–170 (2013)). In that paper, the authors showed that iGluSnFR localizes on cell surface in cultured neurons, but displays intracellular puncta in HEK293 cells and cultured astrocytes. Together, these results for both eLACCO1.1 and iGluSnFR suggest that the specific distribution pattern of membrane localized proteins could depend on cell type.

To more clearly acknowledge the punctate fluorescence of eLACCO1.1 on cultured neurons, we have now added the following sentences to the main text (lines 164–167), “*To characterize the performance of eLACCO1.1 in neurons, we expressed eLACCO1.1 in rat primary cortical neurons. We observed that neurons expressing eLACCO1.1 exhibited bright membrane-localized fluorescence with some puncta apparent (Fig. 4f).*”

- Regarding the slice experiment in Fig 2a-d, given how other sensors operate in slice or in vivo it is not too surprising that some attenuation to the dynamic range is observed. However, it would be helpful to have the authors perspective on what might cause this. In particular, have experiments been done in which slices have been treated with LDH or other lactate degrading enzyme to determine if basal lactate concentration is high in these acute slice preps, which could cause the attenuation in dynamic range?

Thank you very much for helpful comments on the attenuation of the dynamic range of biosensor in brain slice. In previous work, authors of this manuscript (Ciaran Murphy-Royal, Jaideep S. Bains and Grant R. Gordon) have prepared acute brain slices in the same way as in the current work and found that the basal concentration of L-lactate was 0.18 ± 0.05 mM using an enzymatic biosensor (please see the Supplementary Figure 6a, attached below, from the previous publication Murphy-Royal, C. et al. *Nature Communications* **11**, 2014 (2020)). According to the dose-response curve of eLACCO1.1 (Figures 2f and 4e in our manuscript, apparent K_d of 3.9 and 1.6 mM, respectively), this basal L-lactate level should not have much impact on the dynamic range of eLACCO1.1. A similar attenuation of biosensor dynamic range when moving from in vitro to in slice has been reported for a genetically encoded serotonin biosensor iSeroSnFR ($\Delta F/F \sim 9$ (in vitro), 0.8 (in slice), Unger, E. et al. *Cell* **183**, 1–17 (2020)). It is possible that tissue specific post-translational modifications or macromolecular interactions are interfering with the biosensor function in the complex milieu of the acute brain slice.

To describe our perspective on the attenuation of dynamic range of eLACCO1.1 in slice, we have now added the following paragraph to the Discussion section of the main text (lines 278–287), “*As with many genetically encoded biosensors^{8,11}, the maximal fluorescence response of eLACCO1.1 in acute brain slice ($\Delta F/F \sim 0.7$) is smaller than that in HeLa cells ($\Delta F/F \sim 3$) and cultured neurons ($\Delta F/F \sim 2$). A previous study with the extracellular neurotransmitter serotonin biosensor iSeroSnFR similarly reported that its fluorescence response in acute brain slice ($\Delta F/F \sim 0.8$) is smaller than that in HEK293T cells ($\Delta F/F \sim 8$) and cultured neurons ($\Delta F/F \sim 6$) upon treatment with 1 mM serotonin¹¹. In*

addition to the smaller fluorescence response, eLACCO1.1 shows a relatively large variation of fluorescence response in acute brain slice (**Fig. 4j**). To develop a next generation eLACCO variant with improved performance in brain tissues, it might be necessary to combine bacteria-based high-throughput directed evolution with secondary neuron- or slice-based assessments to identify those variants that best retain their performance in brain tissues.”

Reviewer #2

The manuscript presents a new green fluorescent genetically encoded biosensor, eLACCO1.1, that can be targeted to the surface of mammalian cells and used to report changes in extracellular [L-lactate]. It is the first reported sensor of extracellular lactate, and due to the authors’ substantial effort at in vitro directed evolution, it displays large intensity changes in response to lactate. The sensor is specific for lactate over other chemically similar compounds, but it has a requirement for calcium and therefore cannot be used to measure intracellular [lactate].

The authors propose that this sensor should be of use in investigations of cell-to-cell shuttles of L-lactate, such as the astrocyte-neuron lactate shuttle. However, the value of the extracellular sensor for such studies is not completely clear, as a detected change in extracellular [lactate] could not easily help determine the cellular source of lactate (e.g. neurons or astrocytes?) or the directionality of lactate movement between cells. It could certainly be valuable to learn about the changes in extracellular [lactate] in different brain regions under different conditions (as previously done with microdialysis). Both for investigating the shuttle and for a generic characterization of lactate behavior, it would be important to establish that the sensor can be used for quantitative measurement of [lactate] and not just visualization of changes. In principle, this should be possible, as the sensor is ratiometric with different excitation wavelengths; but ratio imaging to obtain calibrated estimates of [lactate] was not done in any of the trial experiments here.

We appreciate your suggestion. To investigate the potential applicability of eLACCO1.1 as a quantitative extracellular L-lactate biosensor, we have now added the ratiometric imaging data to Supplementary Figure 9 and the following text to the main text (lines 153–156), “eLACCO1.1 also exhibited an L-lactate-dependent change in the ratio of excitation at 365 and 470 nm, suggesting eLACCO1.1 could be applicable as both an intensimetric and ratiometric biosensor (**Supplementary Fig. 9**).”

In general, the trial experiments here do not help much to assess the usefulness of the new probe for monitoring physiological changes in lactate. The brain slice experiments simply measure the response of the sensor to bath-applied lactate, rather than to endogenous production and release of lactate. The sizes of these responses are also very variable (Fig. 2d). The glioma responses to altered [glucose] show some change, presumably in endogenously produced lactate, but the manipulation is very large (0 to 25 mM glucose) and the response is quite slow. Also, although the sensor is likely to be fully extracellular, no experiments are done to test this (for instance, insensitivity of the response to blockade of the transporter).

Thank you very much for your comments. To better assess the usefulness of the eLACCO1.1 biosensor, we have now added the imaging data under more physiologically relevant condition of low glucose concentration into the Figure 5d-f and the following sentences to the main text (lines 195–202), “To image the production and export of endogenous L-lactate under a more physiologically relevant condition, we observed eLACCO1.1 expressing T98G cells treated with a physiological plasma concentration of glucose (5.6 mM, **Fig. 5d-f**)²⁰. Treatment with NCI-737, an inhibitor of lactate dehydrogenase (LDH), caused a slight increase in the fluorescence response of deLACCO. In contrast, eLACCO1.1 showed a decrease in the fluorescence response upon NCI-737 treatment. The opposite effects of NCI-737 on the responses of eLACCO1.1 and deLACCO are consistent with the observed fluorescence changes being due to changes in the extracellular L-lactate concentration.”

The slow increase in fluorescence signal in response to production of endogenous L-lactate (Figure 5c,f) is presumably due to the diffusion and dilution of L-lactate secreted into the imaging medium.

To more clearly describe the large variation of $\Delta F/F$ in acute brain slices in the context of potential limitations of eLACCO1.1, we have now added the following paragraph to the Discussion section of the main text (lines 278–287): “As with many genetically encoded biosensors^{8,11}, the maximal fluorescence response of eLACCO1.1 in acute brain slice ($\Delta F/F \sim 0.7$) is smaller than that in HeLa cells ($\Delta F/F \sim 3$) and cultured neurons ($\Delta F/F \sim 2$). A previous study with the extracellular neurotransmitter serotonin

*biosensor iSeroSnFR similarly reported that its fluorescence response in acute brain slice ($\Delta F/F \sim 0.8$) is smaller than that in HEK293T cells ($\Delta F/F \sim 8$) and cultured neurons ($\Delta F/F \sim 6$) upon treatment with 1 mM serotonin¹¹. In addition to the smaller fluorescence response, eLACCO1.1 shows a relatively large variation of fluorescence response in acute brain slice (**Fig. 4j**). To develop a next generation eLACCO variant with improved performance in brain tissues, it might be necessary to combine bacteria-based high-throughput directed evolution with secondary neuron- or slice-based assessments to identify those variants that best retain their performance in brain tissues.”*

To better test the extracellular localization of eLACCO1.1, we have now added results from imaging of $\Delta F/F$ in the presence and absence of a monocarboxylic transporter inhibitor to block lactate flux across the cell membrane (AR-C155858). These results are shown in Figure 3e and the following text has been added to the main text (lines 141–147), *“Bath application of 10 mM L-lactate resulted in no significant difference in $\Delta F/F$ in the presence versus absence of monocarboxylic transporter inhibitor AR-C155858, though the variability in $\Delta F/F$ was greatly decreased by this treatment (**Fig. 3e**). This result suggests that only the fraction of eLACCO1.1 on the cell surface contributes to the fluorescence response associated with changes in the extracellular L-lactate concentration. The decreased variability associated with AR-C155858 treatment is attributed greater consistency in extracellular lactate concentration due to blockage of lactate uptake into cells.”*

The final version of the sensor with an affinity suitable for physiologically-relevant measurements (eLACCO1.1) is missing some important characterizations. Most importantly, its responsiveness to/requirement for Ca^{2+} is not presented. While the prototype sensor, eLACCO1, is reported to require near micromolar Ca^{2+} for lactate responses, the same information is not provided for the mature sensor, eLACCO1.1. The crystal structure shows that calcium is bound in concert with lactate in the binding pocket, so given that the lactate affinity was reduced by 1,000-fold by making a mutation in the binding pocket, one cannot assume that Ca^{2+} affinity remains constant. Given that the two ligands bind together, the $[\text{Ca}^{2+}]$ may also affect the dose-response midpoint for lactate. Likewise, the authors only present binding specificity data (Sup. Fig. 5c) and pH responses (Sup. Fig. 5d) for eLACCO1, but not eLACCO1.1.

To better clarify the effect of Ca^{2+} (and Mg^{2+}) on the functionality of affinity-tuned eLACCO1.1, we have now added the Ca^{2+} (and Mg^{2+}) titration data to the main Figure 2c and the following text to the main text (lines 103–107), *“To investigate the effect of Ca^{2+} on the biosensor functionality, we determined the dependence of the fluorescence intensity of eLACCO1.1 on L-lactate and Ca^{2+} . These experiments revealed that the biosensor only functions as an L-lactate biosensor at concentrations of Ca^{2+} greater than 9 μM (**Fig. 2c**)”.*

The specificity of the affinity-tuned eLACCO1.1 was shown in the original Figure 1e which has now been moved to Figure 2f. To better clarify the pH sensitivity of eLACCO1.1, we have now added the pH titration data in Figure 2e and the following sentence to the main text (lines 116–118), *“The fluorescence of eLACCO1.1 is pH dependent, exhibiting pK_a values of 7.4 and 9.4 in the presence and absence of L-lactate, respectively (**Fig. 2e**)”*

Minor

1. The legend of Fig. 2e does not adequately describe the experiment being performed; there is no mention that glucose is being applied, and it is presumably the glucose that is driving lactate production.

In addition to inhibiting the MCT, phloretin is known to inhibit glucose transporters. A more specific MCT inhibitor (e.g. AR-C155858) would probably have been better.

Thank you for your comments. To better describe the imaging condition using a high concentration of glucose, we have now revised the legend of Figure 5b (originally, Figure 2e) as, “*Representative fluorescence images of T98G cells expressing eLACCO1.1 or deLACCO treated with 25 mM glucose.*”, and revised the main text (lines 188–191) as, “*Upon treatment with a high glucose concentration (25 mM), which is expected to stimulate the production of endogenous L-lactate, eLACCO1.1 on the surface of T98G cells underwent an increase in fluorescence consistent with increased secretion of L-lactate (Fig. 5a-c).*”

To investigate the effect of AR-C155858, we added the imaging data using AR-C155858 under the same high-glucose condition into the Figure 5b,c (originally, Figure 2e,f). Accordingly, we have now revised the main text (lines 191–195) as, “*In the presence of phloretin or AR-C155858, two inhibitors of the monocarboxylate transporter, the glucose-dependent fluorescence increase was diminished. In both the presence and absence of phloretin or AR-C155858, the control biosensor deLACCO showed no substantial change in fluorescence intensity, indicating that the observed fluorescence changes were due to the L-lactate-dependent response of eLACCO1.1.*”

2. In Fig 2d, the variability is enormous, which the authors do not comment on. Given this variability, the “representative traces” shown in Fig. 2c are not sufficient (need to show mean +/- standard error) nor are they representative (the trace in 2c shows a >1.5-fold change in $\Delta F/F$ in response to 10 mM lactate, while the y-axis in 2d only goes to 1.2).

Thank you for your comments. To better describe the large variability of $\Delta F/F$, we have now revised the main text (lines 179–182) as, “*Bath application of L-lactate elicited a variable and significant increase in eLACCO1.1 fluorescence at all doses tested: 1 mM ($\Delta F/F = 0.19 \pm 0.04$), 2.5 mM ($\Delta F/F = 0.48 \pm 0.15$), and 10 mM ($\Delta F/F = 0.65 \pm 0.17$) (Fig. 4h-j)*”. To better describe the fluorescence traces, we have revised Figure 4i (originally, Figure 2c) by showing mean \pm standard error of mean.

3. Main text, line 61: The sentence describing the specificity for lactate refers to Sup Fig. 5d, but it should refer to Sup Fig. 5c.

Thank you for pointing this out. We have fixed this error.

4. The photobleaching and dose-response panels of Supp. Fig. 8 (b and d) have been reversed.

Thank you for pointing this out. We moved the original Supplementary Figure 8b-d (the live cell characterization of eLACCO1.1) to Figure 4 and this unfortunate error has been corrected.

Reviewer #3

This is a very interesting manuscript in which Nasu and colleagues designed and developed a genetically-encoded fluorescent biosensor that detects extracellular L-lactate. The manuscript is nicely

described and elegantly presented, and the experimental work behind the development of this biosensor is very robust, leading to the proposal of a biosensor likely suitable for the assessment of intercellular exchange of L-lactate between, e.g., astrocytes and neurons -although it might be useful for determining L-lactate exchange amongst cancer cells, for instance. Therefore, it seems reasonable that the many researchers using this biosensor should know in more detail to which extension its sensitivity and/or selectivity is/are influenced by pH and calcium, two important factors that enormously change during neurotransmission. Therefore, these issues should be reinforced according to the following comments.

Thank you for your supportive and valuable comments.

1. As the authors are aware, one of the major problems with protein-based biosensors is their vulnerability to protonation making them highly sensitive to pH changes. This is particularly important during physiological neurotransmission, when brain pH may shift between 6.5 and 8.0 [M Chesler & K Kaila (1992) Modulation of pH by neuronal activity. Trends Neurosci 10:396-402. doi: 10.1016/0166-2236(92)90191-a. PMID: 1279865]. The authors indeed aimed to address this issue by determining the normalized fluorescence intensity of eLACCO1 with or without 10 mM L-lactate in vitro (i.e., in the absence of cells) in a wide range of pH (Supplementary Fig. 5d). According to Suppl. Fig, 5d, the largest change in normalized fluorescence intensity took place up to pH value of 7.0 (with L-lactate). However, the experimental values were mathematically adjusted to a sigmoid curve leading to the apparent conclusion that the normalized fluorescence intensity did not change in a pH range between 7 and 10. However, looking at the actual data points, the normalized fluorescence intensity progressively increased as from pH 6.5, i.e. a pH value that is within the physiological pH range during neurotransmission (Chesler & Kaila, 1992). Admittedly, the observed changes in normalized fluorescence intensity between pH values of 7 and 10 are subtle despite progressive suggesting an effect; however, and importantly, in this experiment the fluorescence intensity values were normalized to that obtained at pH 2, hence abrogating the possibility to ascertain how much is the actual change size in absolute fluorescence, particularly in the physiological pH range 6.5-8. Given the critical importance of this issue, especially for the neuroscience community who, according to the authors, will likely be interested in this biosensor, it would be needed (i) to represent the graph of Suppl. Fig. 5d using the raw data (i.e., not the normalized fluorescence), and (ii) given the likely possibility that the biosensor's pH sensitivity may be different in a physiological setting, the authors should also perform a simple experiment that shows the unnormalized fluorescence intensity values in cells expressing eLACCO1 at a fixed L-lactate concentration (e.g., the K_d) in the pH range around that physiologically achievable (i.e., from 6.5 to 8).

To address point (i), we changed the normalized pH titration graph (originally, Supplementary Figure 5d) to an unnormalized one (Supplementary Figure 4d). To better clarify the pH sensitivity of affinity-tuned eLACCO1.1, we have now added the pH titration data in Figure 2e and the following text to the main text (lines 116–118), “*The fluorescence of eLACCO1.1 is pH dependent, exhibiting pK_a values of 7.4 and 9.4 in the presence and absence of L-lactate, respectively (Fig. 2e).*”.

To address point (ii), we have investigated the pH sensitivity of eLACCO1.1 in living cells by recording of fluorescence intensity of eLACCO1.1-expressing HeLa cells in the pH range from 6.5 to 8. We have now added the pH titration data, including unnormalized trace of fluorescence intensity, in Supplementary Figure 11 and revised the main text (lines 160–162) as, “*Cell surface-targeted eLACCO1.1 has an in situ apparent K_d of 1.6 mM (Fig. 4e) and displays Ca²⁺ and pH dependent fluorescence as shown in **Supplementary Figs. 10 and 11.***”

2. Similarly to the previous point, calcium (as magnesium) is an important messenger during neurotransmission and, accordingly, the authors addressed how much the eLACCO1 normalized fluorescence intensity changed with calcium (and magnesium) concentrations. Unfortunately, the normalized data likely avoids the observation of possible subtle changes in fluorescence intensities at calcium concentrations at the synaptic cleft during neurotransmission, which varies from 0 to 2 mM [Cohen JE, Fields RD. (2004) Extracellular calcium depletion in synaptic transmission. *Neuroscientist* 10(1):12-7. doi: 10.1177/1073858403259440. PMID: 14987443]. Moreover, during neurotransmission, a 0 mM calcium concentration is easily achievable, hence it remains unknown whether the L-lactate detection by the eLACCO1 biosensor would be affected. This is also important for cancer research, given the extreme changes in extracellular calcium concentration that can be very high or 0. Therefore, the authors should perform a simple experiment to show the unnormalized fluorescence intensity values in cells expressing eLACCO1 at a fixed L-lactate concentration (e.g., the K_d) in the extracellular calcium range around that achievable physiologically (i.e., from 0 to 2 mM).

We appreciate your suggestion. This point was also brought up by Reviewer #2. We repeat the response here. To better clarify the effect of Ca^{2+} (and Mg^{2+}) on the functionality of affinity-tuned eLACCO1.1, we have now added the unnormalized Ca^{2+} (and Mg^{2+}) titration data in Figure 2c and the following text to the main text (lines 103–107), *“To investigate the effect of Ca^{2+} on the biosensor functionality, we determined the dependence of the fluorescence intensity of eLACCO1.1 on L-lactate and Ca^{2+} . These experiments revealed that the biosensor only functions as an L-lactate biosensor at concentrations of Ca^{2+} greater than 9 μM (Fig. 2c)”*. In addition, we changed the normalized Ca^{2+} titration graph (originally, Supplementary Figure 5a) to one that is not normalized (Supplementary Figure 4a).

To investigate the Ca^{2+} sensitivity of eLACCO1.1 in living cells, we performed the recording of fluorescence intensity of eLACCO1.1-expressing HeLa cells in the Ca^{2+} range from 0 to 2 mM in the presence of 10 mM L-lactate. We have now added the Ca^{2+} titration data in Supplementary Figure 10 and revised the main text (lines 160–162) as, *“Cell surface-targeted eLACCO1.1 has an in situ apparent K_d of 1.6 mM (Fig. 4e) and displays Ca^{2+} and pH dependent fluorescence as shown in **Supplementary Figs. 10 and 11.**”*

Reviewers' Comments:

Reviewer #1:

Remarks to the Author:

In general, Nasu & Campbell and co-workers have addressed all my initial concerns. However, iodoacetate is not uncommonly used to inhibit glycolysis, and therefore I recommend inclusion of the iodoacetate data in the Supplement. The aggregation phenomenon is important to report should users need to troubleshoot issues with oxidation-induced puncta in their own experiments. Otherwise, much of the pertinent information has been moved from the Supplement back into the main text, and the manuscript reads well with the revised text and figures. The description of the protein engineering will provide useful lessons to the sensor community, especially with the addition of the sensor crystal structure. The characterization in solution, on cultured cells, and in slice illustrate the potential for its use mapping extracellular lactate levels.

Reviewer #2:

Remarks to the Author:

The authors have done a thorough job of responding to the earlier remarks, and the paper is now quite sound and significant.

Reviewer #3:

Remarks to the Author:

The authors have adequately addressed all comments raised in the former reviewer's report and, therefore, the manuscript is now improved and this reviewer has no further concerns.

Dear Dr. Eldridge and all reviewers,

We greatly appreciate the additional comments and approval from the reviewers. To address the most recent comments, we have revised our manuscript as described on the following page. The table below summarizes the revised numbering of figures and tables. Newly added figures are shown in blue.

New	Old	Title
Main figures		
Fig. 1	Fig. 1	Development of a genetically encoded L-lactate biosensor, eLACCO1.
Fig. 2	Fig. 2	In vitro characterization of affinity-tuned eLACCO1.1.
Fig. 3	Fig. 3	Targeting of eLACCO1.1 to the extracellular environment.
Fig. 4	Fig. 4a-f	Characterization of eLACCO1.1 in live mammalian cells.
Fig. 5	Fig. 4g-j	Two-photon imaging of L-lactate on astrocytes in acute brain slices.
Fig. 6	Fig. 5	Imaging of endogenous L-lactate release from glioblastoma cells.
Supplementary figures		
Fig. S1	Fig. S1	Construction of the biosensor prototype.
Fig. S2	Fig. S2	Biosensor prototypes based on the various TTHA0766 homologues.
Fig. S3	Fig. S3	Sequence alignment of TTHA0766, cpGFP, and eLACCO1.
Fig. S4	Fig. S4	In vitro characterization of eLACCO1.
Fig. S5	Fig. S5	Crystal structure of eLACCO1.
Fig. S6	Fig. S6	Affinity tuning of eLACCO1.
Fig. S7	Fig. S7	In vitro characterization of deLACCO.
Fig. S8	Fig. S8	Membrane trafficking of eLACCO1.1 with various leader sequences.
Fig. S9	Fig. S9	Ratiometric imaging of eLACCO1.1 on live HeLa cells.
Fig. S10	Fig. S10	Ca ²⁺ titration on live HeLa cells.
Fig. S11	Fig. S11	pH titration on live HeLa cells.
Fig. S12	Fig. S12	Stopped-flow analysis of eLACCO1 and eLACCO1.1.
Fig. S13	Fig. S13	Attempted imaging of extracellular L-lactate with Laconic.
Fig. S14		Attempted imaging of eLACCO1.1-expressing T98G cells treated with iodoacetate.
Tables		
Table 1	Table 1	One- and two-photon photophysical parameters of eLACCO1.1.
Table S1	Table S1	Crystallographic and refinement statistics of eLACCO1.

Reviewer #1

In general, Nasu & Campbell and co-workers have addressed all my initial concerns. However, iodoacetate is not uncommonly used to inhibit glycolysis, and therefore I recommend inclusion of the iodoacetate data in the Supplement. The aggregation phenomenon is important to report should users need to troubleshoot issues with oxidation-induced puncta in their own experiments. Otherwise, much of the pertinent information has been moved from the Supplement back into the main text, and the manuscript reads well with the revised text and figures. The description of the protein engineering will provide useful lessons to the sensor community, especially with the addition of the sensor crystal structure. The characterization in solution, on cultured cells, and in slice illustrate the potential for its use mapping extracellular lactate levels.

Thank you for your supportive comments and helpful suggestion on iodoacetate experiment. To address your suggestion, we have now added the imaging data of eLACCO1.1 expressing T98G cell treated with 100 μ M iodoacetate into the Supplementary Figure 14 and the following sentences to the main text (lines 200–205), *“We first attempted to inhibit production of endogenous L-lactate by treating the cells with 100 μ M iodoacetate, an inhibitor of glyceraldehyde 3-phosphate dehydrogenase (GAPDH). Unexpectedly, this stimulation resulted in apparent aggregation of eLACCO1.1 (Supplementary Fig. 14). To avoid this iodoacetate-induced aggregation artefact, we turned to using NCI-737, an inhibitor of lactate dehydrogenase (LDH) (Fig. 6e,f).”*

Reviewer #2

The authors have done a thorough job of responding to the earlier remarks, and the paper is now quite sound and significant.

Thank you. We appreciate all of your comments on the revision of this manuscript.

Reviewer #3

The authors have adequately addressed all comments raised in the former reviewer's report and, therefore, the manuscript is now improved and this reviewer has no further concerns.

Thank you. We are grateful for all of your suggestions on the revision of this manuscript.